# Multimodal communication and audience directedness in the greeting behaviour of semi-captive African savannah elephants
Vesta Eleuteri [1] ✉, Lucy Bates [2], Jake Rendle-Worthington[3], Catherine Hobaiter [4,6] & Angela Stoeger [1,5,6] ✉

Many species communicate by combining signals into multimodal combinations. Elephants live in multi-level societies where individuals regularly separate and reunite. Upon reunion, elephants often engage in elaborate greeting rituals, where they use vocalisations and body acts produced with different body parts and of various sensory modalities (e.g., audible, tactile). However, whether these body acts represent communicative gestures and whether elephants combine vocalisations and gestures during greeting is still unknown. Here we use separation-reunion events to explore the greeting behaviour of semi-captive elephants *(Loxodonta africana)*. We investigate whether elephants use silent-visual, audible, and tactile gestures directing them at their audience based on their state of visual attention and how they combine these gestures with vocalisations during greeting. We show that elephants select gesture modality appropriately according to their audience's visual attention, suggesting evidence of first-order intentional communicative use. We further show that elephants integrate vocalisations and gestures into different combinations and orders. The most frequent combination consists of rumble vocalisations with ear-flapping gestures, used most often between females. By showing that a species evolutionarily distant to our own primate lineage shows sensitivity to their audience's visual attention in their gesturing and combines gestures with vocalisations, our study advances our understanding of the emergence of first-order intentionality and multimodal communication across taxa.

Animals communicate using vocalisations, gestures, facial expressions, scent cues, and other signals conveying information across different sensory modalities. Research on animal communication has focused on exploring signal types or modalities separately[1,2]. However, isolating a signal from its broader communicative context can limit our understanding of its function. For example, in human speech, we frequently use paralinguistic visual signals, such as facial expressions or gestures, to adjust the meaning of messages[2].

Multimodal signalling is discriminated based on the production or perception of signals: multicomponent combinations involve signals from different articulators (i.e., production organs), such as vocalisations produced with the vocal tract or gestures with the limbs, while multisensory combinations involve signals conveying sensory information (e.g., visual and acoustic) perceived through different sensory channels (e.g., vision and hearing)[3,4]. Multicomponent and multisensory combinations may serve different communicative functions. In redundant combinations, signals convey the same information to facilitate its detection or enhance recipient response, while in non-redundant combinations the functions of component signals may be refined evoking a distinct response by recipients[5,6]. Outside of apes, research has focused on multisensory combinations during contexts including courtship (crickets[7]; birds[8]), agonistic interactions (frogs[9]; Cichlid fish[10]), and antipredator displays (insects[11]; squirrels[12]). For example, flies court by combining visual, acoustic, vibratory, and chemical signals[13], while different bird species by combining elaborate visual displays

[1]Department of Behavioral & Cognitive Biology, University of Vienna, Vienna, Austria. [2]Department of Psychology, University of Portsmouth, Portsmouth, UK. [3]Elephant CREW, Jafuta Reserve, Victoria Falls, Zimbabwe. [4]School of Psychology and Neuroscience, University of St Andrews, St Andrews, UK. [5]Acoustic Research Institute, Austrian Academy of Sciences, Vienna, Austria. [6]These authors jointly supervised this work : Catherine Hobaiter, Angela Stoeger.
✉e-mail: vesta.eleuteri@univie.ac.at; angela.stoeger-horwath@univie.ac.at

with songs[8]. Conversely, research on primates has focused on multi-component combinations showing some evidence of refinement[14]. For example, bonobos combine gestures with different facial expressions to convey aggressive or playful intentions, while in chimpanzees vocal-gestural combinations may serve as attention getters to gestures or to disambiguate their meaning[15–17].

African savannah elephants (*Loxodonta africana*) represent a promising candidate to further explore the use and functions of multimodal communication. They possess a rich communication system including acoustic, chemical, seismic, visual, and tactile signals often combined in elaborate displays[18–21]. The most frequent call is the rumble, which contains information like individual identity[22,23], sex[24], age[25], reproductive state[26,27], and arousal[28]. Other common vocalisations during high arousal are trumpets and roars[21,29,30]. Additionally, the mouth, temporal glands, and genitals of elephants produce chemical secretions containing information relating to individual identity, reproductive state, or emotional state[19], and elephants frequently investigate these body parts or their secretions with their trunks[20]. Lastly, elephants use many visual and tactile body acts across different behavioural contexts, suggesting vision and touch are particularly important in their communication[30]. However, it is not yet known if or to what extent elephants use these body acts to communicate flexibly and intentionally.

Intentionality is argued to underly the uniqueness of human language[31]. With language we choose to communicate an underlying thought to a partner, by taking into account their mental states[32,33]. The capacity for intentional communication can be distinguished into different levels[34]. Zero-order intentionality is attributed to signals triggered as simple reactions to stimuli, whereas first-order (or goal-directed) intentionality is attributed to signals produced to communicate goals and elicit behavioural reactions in recipients, and second-order intentionality to signals produced to change the mental states of recipients[34]. Most animal signals are suggested to be zero-order intentional[31,35,36], and there is scarce evidence of second-order intentionality outside of humans (but see ref. [37,38]). But first-order intentionality—an essential precursor to second-order intentionality—has been found to be widespread in other apes[39]. All apes use large repertoires of gestures to flexibly communicate goals, acquiring this capacity during infancy, similarly to the onset of speech in human infants[40,41]. One aspect of first-order intentionality is that signals are directed towards an audience: the signaller must be in the presence of a recipient, visually check the recipient, and show sensitivity to the recipient's attentional state by selecting the appropriate modality of the signal (i.e., audience directedness)[42,43].

Communicative capacities are adapted to the socio-ecological needs and sensory-motor systems of the species employing them. Other apes share our body plan and richly structured social lives, but to understand which social and ecological factors led to the evolution of first-order intentionality we need to also explore species more distantly related to us, including those with different sensory-motor systems[44]. Elephants are particularly good candidates because they are physically different and evolutionarily distant from us[45], but share our long lives and multi-level fission-fusion societies[46–48] in which goal-directed flexible gesturing might mediate diverse social interactions. Moreover, wild elephants perform many body acts across different behavioural and social contexts[30], and semi-captive elephants have been shown to adjust their use of visual gestures towards a human experimenter depending on her state of visual attention, suggesting evidence of first-order intentional use[49]. However, whether elephants direct gestures at conspecifics and which, if any, of their described body acts may represent first-order intentional gestures has not been systematically investigated.

Like other species living in fission-fusion societies, elephants regularly separate and reunite[48], and may engage in greetings upon reunion. Greetings in animals are suggested to function to mediate social interactions[50,51] by reducing tension and avoiding conflict[52], reaffirming existing social bonds[53], re-establishing dominance dynamics[54], or providing updated information on individuals such as reproductive or affective states[55]. Several species produce and integrate signals when greeting group members[52,56,57], as do elephants. When they meet, wild male elephants may occasionally rumble and often direct their trunks to their conspecifics' mouth, temporal glands,

or genitals, apparently to facilitate positive interactions or gain socially relevant olfactory information[19,30,55]. Moreover, while unknown captive female elephants do not greet upon meeting, related or closely bonded wild and captive female elephants engage in elaborate behavioural displays apparently to facilitate recognition and strengthen social bonds[20,30]. Females may vocalise with rumbles, roars, and trumpets; spread, and flap their ears; touch each other; back into each other; and defecate or urinate[30]. The diversity of body acts and vocalisations makes greeting a promising context in which to explore audience-directed gestures and multicomponent combinations in elephant communication.

We explored the use of audience-directed gestures and multi-component combinations of vocalisations and gestures during greetings in a group of semi-captive African savannah elephants. Firstly, we identified the range of vocalisations and body acts used. Secondly, we described which, if any, body acts represent audience-directed gestures by investigating whether elephants target body acts at recipients selecting their modality appropriately according to recipients' visual attentional states. We predicted that elephants would use more audible or tactile body acts when recipients are not attending and more silent-visual body acts when they are attending[41,58]. We also explored whether elephants combine specific vocalisations (e.g., Rumble) with specific gestures or body acts (e.g., Ear-Flapping) and whether they would order them in specific ways within multicomponent combinations. Because elephant greetings are described as chaotic combinations of signals[20,30], we predicted that specific signals would not be ordered when combined together in this context.

Finally, to understand the functions of multicomponent combinations during greeting, we investigated whether individual and social factors affected the combined use of vocalisations with gestures or body acts. Because of their more elaborate greetings in the wild[30], we predicted that females would greet with common multicomponent combinations more frequently than males. Moreover, we predicted an order effect in the general use of vocalisations and gestures in combination: elephants would first vocalise and then start gesturing, possibly to elicit the attention of recipients to gestures to enhance the transmission of information[3,5].

We found that elephants greet with specific vocalisations and body acts. Most body acts were targeted at recipients and their modality was selected appropriately according to the recipient's state of visual attention, thus representing audience-directed gestures. Elephants combined specific vocalisations with gestures or body acts in specific multicomponent combinations and Rumble-Ear-Flapping was the most frequent combination, especially used among females. By showing that elephants produce audience-directed gestures and specific multicomponent combinations during greeting, our study expands the current knowledge on elephant communication and enhances our understanding of the evolution of first-order intentional communication and signal integration beyond the primate lineage.

## Methods
### Study site and subjects
Separation-reunion procedures are a useful way to promote greeting behaviour and study vocalisations in elephants[59]. While wild elephants are not used to being separated by humans and show high levels of stress if forced apart[60], semi-captive elephants are regularly separated for short periods for training purposes, interactions with humans, or medical interventions. Moreover, recognising who is vocalising is extremely challenging in wild elephants, especially during their elaborate greeting behaviour where individuals often call together[20,30]. Therefore, we decided to conduct our study of elephant greeting behaviour on a group of semi-captive elephants.

We collected data in November and December 2021 with a group of semi-captive African savannah elephants in the Jafuta Reserve in Zimbabwe. The reserve consists of teak forest and grassland and is inhabited by other species, including wild elephants. The semi-captive group consists of 9 elephants (four males and five females) engaged by elephantCREW in non-invasive interactions with tourists and locals.

The STRANGE framework was established to identify sampling biases in studies of animal behviour that may affect reproducibility and generalisability of findings[61,62]. Originating from different wild herds and being in semi-captivity, our study subjects live in an artificial social group comprising adult males and females, which may impact their social behaviour and communication. In line with the STRANGE framework, we discuss our results by considering the sampling bias in our study[61,62].

## Data collection

Data on elephant greetings were collected from an elevated observation deck positioned next to a water hole, with clear visibility of the elephants for at least 200 m on all sides. To promote greetings between elephants we used a separation-reunion procedure. First, the elephant carers led two elephants more than 200 m apart from each other and behind vegetation so that they were no longer visible to one another. Then, after ten minutes, they led the elephants out of the vegetation and allowed them to freely approach each other. Individual elephants were only separated from each other once per day, and a maximum of two separation trials were conducted each day to minimise any stress to the elephants. Protocols were established to immediately reunite any elephants who showed signs of distress during the separation, but this was never observed so no trials had to be stopped for this reason.

Previous studies described elephants with stronger social bonds engaging in more elaborate greeting behaviour[20,63]. When we separated and reunited individuals with weak social bonds, they did not approach each other and greet. Thus, to have a sufficient sample size of greeting signals per subject, we selected for the separation-reunions 8 pairs of 6 elephants showing strong social bonds, ensuring that each subject had at least two possible partners. To determine the elephants' social bonds, we assessed the association level of each elephant dyad in the group by calculating their Nearest-neighbour index $(NN_{AB})$[64] using the field site's focal data on associations between all elephants collected in the previous year.

The elephant carers collected nearest neighbour data by conducting all-day focal follows of each elephant twice a month, in which they noted the activity and nearest neighbour of the focal individual every 15 minutes. $NN_{AB}$ represents the rate at which an individual B was the closest to the focal individual A during the focal sample. We determined this index using the 15-min scan samples for each individual B to calculate the following proportion:

$$NN_{AB} = \frac{A_f B_{nn}}{A_h}$$

$A_f B_{nn}$ is the number of 15-min scans in which an individual B was the nearest neighbour to the focal individual A when the focal individual A was being followed. $A_h$ is the total number of 15-min scans for A for the period October 2020 to Novermber 2021. We considered strong associates those individuals B who had $NN_{AB}$ indexes greater than one-quarter standard deviation above the mean of the $NN_{AB}$ indexes of all individuals B (Supplementary Data 1, Supplementary Table 1).

We collected video and audio recordings of the reunion event using a Panasonic AG-UX90EJ8 video camera and an omnidirectional Neumann microphone KM183 modified to record frequencies below 20 Hz (flat recording down to 5 Hz) and connected to a Mix Pre-6 sound device recorder at 48 kHz sampling rate at 16-bit amplitude resolution. We transferred videos and audio recordings to a MacBook Pro and synchronised the separate video and audio files using DaVinci Resolve version 17.

## Definitions of signal types and modalities

We focused on signal production and defined multicomponent combinations as the simultaneous or overlapping occurrence of vocalisations with body acts that included visual, tactile, audible, and possible olfactory components. The core unit for the identification of body acts in our repertoire is the action, the physical movement that uniquely characterises a body act (e.g., Reach for extending a body part towards the recipient)[65]. We then

defined body act types as instances of ineffective movement of a body part in a social context that were not used to perform locomotion, foraging, drinking, or self-directed activities (e.g., Trunk-Reach for reaching with trunk). To define different body act types we explored the ethogram of elephant behaviour available on ElephantVoices[30].

As the purpose of greeting is not to elicit a particular response from the recipient that satisfies the signaller, we determined first-order intentional use based on evidence that body acts were targeted towards the audience and chosen appropriately according to the audience's ability to perceive them. Specifically, we defined those body act types that met the first-order intentionality criterion of audience directedness as audience-directed gestures[36]. We required that: (1) the signaller performed the body act type in the presence of the recipient, for example when the recipient emerged from the vegetation and approached the signaller (i.e., social use); (2) the signaller produced the body act type above chance when showing visual attention to the recipient (i.e., audience checking); (3) the signaller manifested sensitivity to the recipient's visual attentional state when selecting the body act modality[42] (See below for details on statistical analyses).

As all potential gestures have a visual component and can, therefore, be visually perceived, we classified body act types by the presence of any additional sensory modality (following[41,58]). When the performance of a body act type did not involve an audible or tactile component in its production it was classified as silent-visual; when it involved an intrinsic audible component it was classified as audible; and when it incorporated physical contact with the recipient it was classified as tactile. Furthermore, due to the described use of olfactory behaviours in elephant greetings[30], we defined olfaction as an additional potential category of modality for body act types that might be used to facilitate exchange of olfactory information (See Table 1 below). In our data, body act types produced with the tail (i.e., tail body act types) except Tail-Touch did not include audible and tactile information so they would be a priori classified as silent-visual. However, preliminary analyses suggested that these were consistently produced irrespective of the recipient's visual attention. Thus, they could be either a) non-intentional body movements, or b) gestures used to convey olfactory information from the genital area to the recipient.

## Data coding

We coded the synchronised videos with the video coding software Elan 6.2 and the audio coding software PRAAT 6.1.54. After the handlers left the subjects, we considered the greeting behaviour to start when one of the subjects produced the first vocalisation and/or body act after they started approaching each other. We considered the greeting to end when both subjects stopped signalling and rested or engaged in another behaviour (e.g., travelling). We annotated all vocalisations and body acts produced during reunions. We coded the videos at three levels of detail: i) the communication event, ii) the vocalisations produced, and iii) the body acts produced. Our coding method was based on the GesturalOrigins bottom-up coding approach[65]. The communication event included information on the greeting communication: signaller and recipient identity, signaller prior and post communication context, use of olfactory behaviours by the signaller. We then coded information on all vocalisations produced by the signaller (e.g., vocalisation type). Lastly, we coded information on all body acts produced by the signaller: body act type, the states of visual attention of the signaller and of the recipient at the onset of production of the body act, and the distance of the recipient from the signaller (See Supplementary Table 2). The average length of our elephants' bodies was 3 m. We established the distance between the subjects by estimating the number of body lengths between them at the onset of production of any body act. Moreover, the position of their eyes on the sides of their head means that elephants have a potential visual field of 313 degrees[66]. However, to be conservative, we considered visual attention to occur when signallers were facing the recipient, were standing at a 90-degree angle to the recipient's eyes, or were looking back at the recipient and the ears were not obstructing their view. Similarly, we considered visual attention to occur in recipients when they were front facing the signaller, their head was aligned at a 90-degree angle to the

**Table 1 | Definitions of all identified signal types and modalities with frequencies of production in our subjects and whether the signals were previously described as part of elephant greeting behaviour in the ElephantVoices ethogram of elephant behaviour[30] (For a video and descriptions of a greeting event see Supplementary Movie 1 and Supplementary Note 1, 2)**

| Signal type[a] | Signal | Modality | Frequency | Definition | Previously described | ElephantVoices Definition |
|---|---|---|---|---|---|---|
| Back-Towards | Gesture | Silent-visual | 18 | Turning rump to recipient and walking backwards towards them. | Yes | Back-Toward |
| Ear-Brush | Gesture | Tactile | 5 | Brushing an ear against a recipient's body part (i.e., ear, head, trunk, rump). | Yes | Ear-Brush |
| Ear-Flapping | Gesture | Audible | 272 | Flapping the ear(s) back and forth. The flapping of the ear(s) produces a distinct sound. | Yes | Rapid-Ear-Flapping |
| Ear-Slap | Gesture | Audible | 15 | Slapping the ear(s) loudly and sharply against the neck and shoulders. | Yes | Ear-Slap |
| Ear-Slight-Spread | Gesture | Silent-visual | 65 | Opening the ear(s) between 45° and 90°. Ear(s) may be lifted as well. | Yes | Ear-Stiff |
| Ear-Spread | Gesture | Silent-visual | 99 | Opening the ear(s) fully at a minimum of ~90° from body. Ear(s) may be lifted as well. | Yes | Ear-Spreading |
| Ears-Stiff | Gesture | Silent-visual | 149 | Opening the ear(s) between 30° and 45°. Ear(s) may be lifted as well. | Yes | Ear-Stiff |
| Head-Raise | Gesture | Silent-visual | 29 | Raising the head extending the neck upwards and outwards. | Yes | Head-Raising |
| Roar | Vocalisation | | 16 | Graded sounds ranging from tonal to chaotic. Usually emitted during distress or high arousal situations[21]. | Yes | Roar |
| Rubbing-Other | Gesture | Tactile | 4 | Rubbing a body part on the recipient. Produced with the head and face. Contact was made with the recipient's ear, head, trunk, or face. | Yes | Social-Rubbing |
| Rumble | Vocalisation | | 227 | Harmonically rich and voiced sounds with fundamental frequencies near the infrasonic range. Emitted for social functions like spatial dynamics and movement coordination or contact calling[18]. | Yes | Rumble |
| Rump-Present | Gesture | Silent-visual | 18 | Presenting own rump to the recipient. | No | Rump-Present |
| Tail-on-Side | Body act | Olfactory? | 51 | Holding the tail out to the right or left of own rump. | No | Previously identified as Tail-Raising |
| Tail-Raise | Body act | Olfactory? | 47 | Raising the tail to more than 90° from the vertical line at rest. | Yes | Tail-Raising |
| Tail-Stiff | Body act | Olfactory? | 40 | Raising the tail to less than 90° from the vertical line at rest. | No | Previously identified as Tail-Raising |
| Tail-Touch | Gesture | Tactile | 10 | Extending the tail to touch the recipient. Contact was made with the recipient's front of body, central body, front limb, or tusk. | No | |
| Tail-Waggling | Body act | Olfactory? | 107 | Waggling the tail raising it up and down from side to side. | Yes | Previously identified as Tail-Raising |
| Trumpet | Vocalisation | | 25 | Loud and higher frequency sounds produced by a forceful expulsion of air from the trunk[21]. Usually emitted during distress or high arousal situations[18]. | Yes | Trumpet |
| Trunk-Reach | Gesture | Silent-visual | 40 | Extending the trunk towards the recipient. May be action of sniffing but suggested to be used as a visual signal[30]. Directed towards the recipient's front of body, central body, genitals, head, mouth, penis, rump, or temporal gland. | Yes | Reach-Touch |
| Trunk-Reach_Touch_Unc | Gesture | | 10 | Same as Trunk-Reach but unclear whether contact with recipient was made. Directed towards the recipient's front of body, central body, head, mouth, penis, or temporal gland. | Yes | Reach-Touch |
| Trunk-Shaking | Gesture | Silent-visual | 15 | Shaking the trunk quickly and repeatedly in different directions. | No | |
| Trunk-Side-Swinging | Gesture | Silent-visual | 10 | Swinging the trunk repeatedly from side to side going up and down on each side. | No | |
| Trunk-Swinging | Gesture | Silent-visual | 10 | Swinging the trunk repeatedly back and forth. | No | |

[a]Gestures/Body acts ending with "ing" indicate repetitive movement of a physical action (e.g., Ear-Flapping involves repeated back and forth movement of the ear(s)).

signaller, or they were looking back at the signaller with the ears not obstructing their view.

Inter-observer reliability was conducted on a subset of 100 signals coded by V.E and a trained coder, Mounia Kehy, on three variables: the signal type used (i.e., body act or vocalisation type), the state of visual attention of the recipient, and the state of visual attention of the recipient (attending = Yes, not attending = No). We found substantial to almost perfect levels of agreement on all three variables (Cohen's kappa: Signal record $K = 0.88$; Signaller Gaze $K = 0.80$; Recipient Visual Attention $K = 0.88$).

## Ethical statement

We have complied with all relevant ethical regulations for animal use. Data collection followed the ASAB guidelines for the treatment of animals during behavioural studies (2018) and the ASAB guidelines for the treatment of animals in behavioural research and teaching (ANIMAL BEHAVIOUR, 135, I-X). Ethical approval for the study was given by the Faculty of Life Sciences of the University of Vienna (Ethical Approval No.2021-021).

## Statistics and reproducibility

To describe the greeting repertoire of our study elephants, we retained all vocalisation types and body act types observed at least twice in at least two elephants during reunion to ensure the signals were representative of the group repertoire (following ref. [58]). The visual acuity of elephants over large distances is not documented. However, as elephants can see a 2.75 cm object at the tip of their 2 m trunk[67], we calculated that they would be able to detect a 2.5–3 m object, which represents the average height of our elephants, from around 180 m away. Similarly, we calculated that they could detect a 1 m object, which represents the minimum size of the body parts used to produce the body acts (e.g., tail, trunk), from around 70 m away. Thus, to be conservative, we excluded cases where the signaller produced a body act when the recipient was more than 100 m away. We further excluded cases where we were unsure whether the signaller was aware of the presence of the recipient. To explore the size of the greeting repertoire, we calculated the number of signal types used in relation to the number of signal cases coded and plotted the data to visually inspect whether the repertoire reached asymptote.

To consider body act types as audience-directed gestures, firstly we required that signallers visually checked the recipient at greater than chance frequency when producing them. Because with tactile body acts signallers can convey information to the recipients without needing to check them visually beforehand, we did not require tactile body act types to meet this condition. We also required that signallers produced silent-visual body act types at the above chance frequency when the recipient was visually attending them. This requirement was not applied to audible or tactile body act types because the recipient does not need to be visually attending them to receive audible or tactile information.

Then, to further investigate whether elephants choose their body act modality appropriately according to the recipient's state of visual attention, we used a Multinomial logit model[68]. The response variable indicated the modality of the body act (i.e., silent-visual, tactile, audible) and the predictor variable indicated whether the body act was produced when recipients were visually attending or not. Because the signals were collected from the same individuals, to avoid pseudo-replication, we fitted the identity of the signaller as a random effect. We included the theoretically identifiable random slope for Signal modality within Signaller.

Additionally, following Hobaiter & Byrne[58], to explore active adjustment of body act modality according to the state of visual attention of the recipient, we calculated the percentage deviation of silent-visual, audible, and tactile body acts according to recipient visual attention. To do so, we first calculated the proportion of silent-visual, audible, and tactile body acts in the entire repertoire. We then separated instances where the recipients were visually attending from those where they were not. Within these two subsets, we calculated the proportion of use of silent-visual, audible, and tactile body acts. Finally, we calculated the percentage deviation in each body act modality when recipients were attending or not using the equation: ($\beta$/

$\alpha - 1) \times 100$. $\beta$ represents the proportion of body acts of a specific modality (e.g., silent-visual) when recipients were in the specific state of visual attention (e.g., visually attending) and $\alpha$ the proportion of body acts in that modality in the entire repertoire. Positive deviations indicate that signallers actively adjust their body acts' modality to the recipient's state of visual attention (Supplementary Data 2).

We restricted analyses to those individuals who contributed at least one body act in each modality and excluded cases where the state of visual attention of the recipient was unclear. The final sample consisted of $n = 670$ body act cases.

Preliminary analyses revealed that tail body act types that could be a priori defined as silent-visual (i.e., Tail-on-Side, Tail-Raise, Tail-Stiff, and Tail-Waggling) were not used above chance when recipients were visually attending. Thus, to further assess whether these body act types could be gestures used for visual communication, we used a Generalised Linear Mixed Model (GLMM) with a binomial error structure and logit link function. The response variable indicated whether the body act was one of the tail body act types or a silent-visual body act (Yes = 1, No = 0) and the predictor variable indicated whether the recipient was attending when the body act was produced (Yes = 1, No = 0). Again, due to pseudo-replication, we fitted the identity of the signaller as a random effect. We included the theoretically identifiable random slope for Recipient visual attention within Signaller. Then, following Hobaiter & Byrne[58], we calculated the percentage deviation of the use of these tail body act types in relation to the recipient's state of visual attention compared to the normal distribution of the established silent-visual body acts (Supplementary Data 3). We excluded cases where the state of visual attention of the recipient was unclear. Doing so resulted in a total of $n = 543$ body act cases included in the analyses. Lastly, we explored the distances at which these tail body act types were produced to understand any alternative functions.

To explore whether elephants produce specific types of multi-component combinations and orders during greeting, we compared the rates of occurrence of combinations of vocalisation and body act types using the linguistic method of Collocation analysis[69]. Collocation analysis is used to identify non-random word combinations in languages by comparing the frequency of co-occurrence of two specific words termed bigrams. Specifically, Multiple Distinctive Collocation Analysis (MDCA) explores the relative attraction (i.e., rate of co-occurrence) between signals within bigrams using one-tailed exact binomial tests on each possible bigram combination providing an estimate of the attraction of signals with each other. MDCA also allows to explore whether signals are used with a particular order within combinations. Because we were interested in whether elephants combine different signal types simultaneously, we ran this analysis on vocalisations and body acts where the durations overlapped. We excluded cases where the start timings of signals were unclear. To fully capture both the order and ways elephants combine vocalisation and body act types we conducted two separate MDCAs. A first MDCA was used to explore the order in which vocalisation and body act types are combined together. The start time of a signal following a signal produced with the same body part depends on the end time of the latter, affecting our ability to detect the order in which vocalisation and body act types are combined together. Thus, if a vocalisation A overlapped with more than one body act produced with the same body part, we included only the first body act overlapping with A (e.g., if Rumble overlapped with Ear-Spread and then Ears-Stiff, we included only Ear-Spread). A second MDCA was used to explore the general frequency of co-occurrence of vocalisation and body act types without taking into consideration any order pattern. Here we included the entire dataset of multiple co-occurring signals. The samples consisted of $n = 337$ distinct bigrams of vocalisations and body acts in MDCA1 and $n = 403$ in MDCA2. Collocation analyses were conducted using R scripts developed by Gries[70].

To understand whether the elephants' use of multicomponent combinations during greeting depends on individual and social factors, we fitted two GLMMs[71] with a binomial error structure and logit link function. In the first GLMM we explored whether individual and social factors affected the order of vocalisations with gestures or body acts in multicomponent

combinations in general. In the second GLMM we explored whether individual and social factors affected the use of the frequent combinations of the vocalisation Rumble with the body act Ear-Flapping. In the first GLMM the response variable indicated whether the combination started with a vocalisation (1) or a gesture/body act (0). In the second GLMM the response variable indicated whether the combination consisted of Rumble with Ear-Flapping in any order (1) or of any other combination (0). In both models the predictors were: the interaction between the sex of the signaller and the type of sex dyad between the signaller and the recipient (i.e., same sex or different sex); and the strength of relationship between the signaller and recipient, for which we used their z-transformed average Nearest-neighbour index (See Supplementary Table 1). Because the samples of all models were composed of signals collected from the same individuals and from the same communication events, to avoid pseudo-replication, we fitted the identity of the signaller as well as the communication number as random effects. To keep the Type I error rate at 5% nominal level, we built maximal models in which we included all theoretically identifiable random slopes[72,73]. We included random slopes for Nearest-Neighbour index and for the interaction between Signaller sex and Sex dyad within Signaller, which was first dummy coded and centred.

To explore the overall effect of the fixed effects, we used a likelihood ratio test comparing the full model with the reduced model without the fixed effects but including the control fixed effects and random effects[74]. We checked for multicollinearity using Variance Inflation Factors[75]. In both models the fixed effects had VIFs close to 1.0. We assessed model stability by comparing the full model estimates with those from models from which random effects were removed one at a time[76]. The first GLMM was fairly stable with respect to the interaction between Signaller sex and Sex dyad (estimate = 0.383; model stability estimates: min = 0.301, max = 0.574). The second GLMM was unstable with respect to the average Nearest-neighbour index and fairly stable with respect to the interaction between Signaller sex and Sex dyad. We present the bootstrapped 95% confidence intervals. In both models the sample consisted of $n = 337$ multicomponent combinations of vocalisations and body acts collected from the 6 elephants.

### Reporting summary
Further information on research design is available in the Nature Portfolio Reporting Summary linked to this article.

## Results
### What signals do elephants use during greeting?
We recorded a total of 1282 signal cases produced by at least two elephants a minimum of two times during 89 greeting events. Of these, 1014 were body acts and 268 vocalisations. Among these, we identified a total of 20 body act types and three vocalisation types. All signal types except Rubbing-Other were used by both males and females (Figs. 1, 2, Table 1). Examination of the cumulative frequency of the signal types revealed that the repertoire reaches asymptote (See Supplementary Fig. 1), suggesting that further observation would unlikely result in the identification of substantial numbers of new signal types.

Out of 89 greeting communication events, 71% involved the use of the olfactory behaviours Urination, Defecation, and Temporal gland secretions, while 24% showed no olfactory behaviours and in 6% it was unknown whether elephants urinated, defecated, or secreted from temporal glands (See Supplementary Table 3).

### Do elephants produce audience-directed gestures during greeting?
Except for the tactile body act type Tail-Touch, all body act types were produced above chance by signallers when they were visually attending the recipient (See Supplementary Fig. 2 and Supplementary Table 4). Most body act types were also produced above chance when the recipient was visually attending them, except body act types produced with the tail (i.e., tail body act types, Fig. 3, See Supplementary Table 5). We observed that 86% ($n = 344/398$) of silent-visual body acts and 83% ($n = 209/253$) of audible body acts were used when the recipient was visually attending. In contrast, only 58% ($n = 11/19$) of tactile body acts were used when the recipient was visually attending. The Multinomial logit model revealed that signallers used audible body acts and silent-visual body acts more often than tactile body acts when recipients were visually attending them (Table 2).

Moreover, tactile body acts showed striking variation in use, decreasing when recipients showed visual attention and increasing when they did not. There was also a decrease in the use of silent-visual body acts when the recipient did not show visual attention. Audible body acts showed a slight increase in the absence of visual attention (Fig. 4).

The GLMM used to assess whether Tail-on-Side, Tail-Raise, Tail-Stiff, and Tail-Waggling could be gestures used for visual communication revealed that Recipient visual attention affected their use by elephants ($\chi^2_1 = 11.025$, $P = 0.001$). Specifically, these tail body act types occurred 13% less often (estimate = −1.858) when recipients were visually attending compared to established silent-visual body acts (Supplementary Table 6). The model explained a low proportion of variance (marginal $R^2 = 0.010$). Moreover, these tail body act types showed no percentage variation in use according to the recipient's state of visual attention, and were actually less likely to be selected when the recipient was attending compared to silent-visual body acts (Supplementary Fig. 3). These results suggest that these tail body act types do not appear to be used in a way that is sensitive to the recipient's ability to perceive them visually. Lastly, we found that, compared to the other tail body act types, Tail-on-Side and Tail-Raise were produced within 1 m from the recipient in most cases (Supplementary Fig. 4). As these results indicate that elephants do not adjust Tail-on-Side, Tail-Raise, Tail-Stiff, and Tail-Waggling to the recipient's ability to perceive the visual information within them, we excluded them as audience-directed gestures (but see Discussion on their potential for exchange of olfactory information).

### What types of multicomponent combinations do elephants produce during greeting?
In MDCA1, 20% ($n = 69$) of the 337 bigrams consisted of combinations of the gesture type Ear-Flapping and the vocalisation type Rumble. The highest

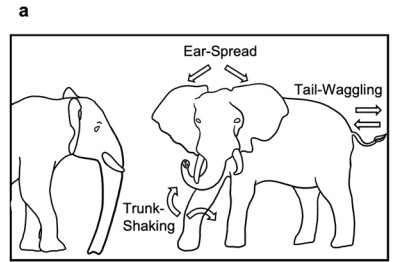
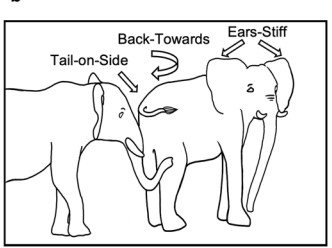
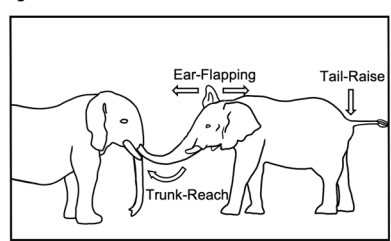

**Fig. 1 | Illustrations of frequent body act types used by semi-captive African savannah elephants during greeting.** The signaller (right) is displayed using different body act types in the panels: **a** Ear-Spread, Tail-Waggling, and Trunk-Shaking; **b** Ears-Stiff, Back-Towards, and Tail-on-Side; **c** Ear-Flapping, Trunk-Reach, and Tail-Raise. For definitions of the body act types see Table 1. Illustrations were drawn by Megan Pacifici.

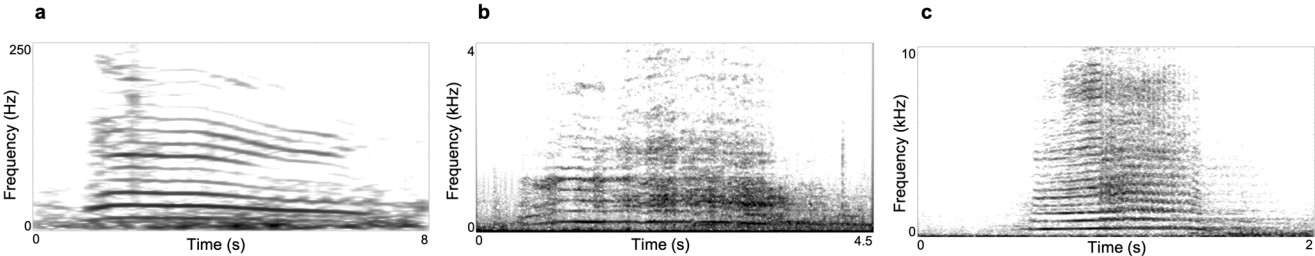

**Fig. 2 | Spectrograms of vocalisation types used by semi-captive African savannah elephants during greeting. a** Spectrogram of a rumble by the male elephant Doma; **b** Spectrogram of a roar by the female Hwange; **c** Spectrogram of a trumpet by Doma. For definitions of the vocalisation types see Table 1.

**Fig. 3 | Percentage of use of body act types where the recipient showed visual attention to the signaller's body act or not at its onset of production during greeting.** "Yes" indicates that the recipient was visually attending the body act; "No" indicates that the recipient was not visually attending the body act. The letters preceding the body act names indicate the body act modality: A = Audible; S = Silent-visual; T = Tactile; U = Unknown (e.g., "A-Ear-Flapping"; S-Back-Towards; T-Ear-Brush; U-Tail-on-Side). n = 910 body acts.

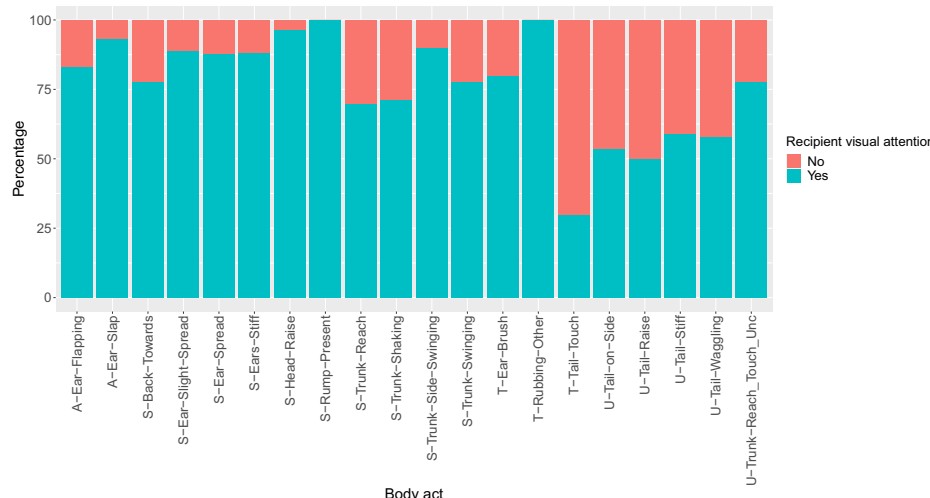

**Table 2 | Results of the Multinomial logit model exploring if the modality of body acts used by semi-captive elephants during greeting varies according to the recipient's state of visual attention**

| | b | SE | z | P | Upr CI | Lwr CI |
|---|---|---|---|---|---|---|
| Modality_Audible ~ (Intercept) | 5.199 | 0.397 | 4.148 | (1) | 2.384 | 11.339 |
| modality_Silent-visual ~ (Intercept) | 6.692 | 0.389 | 4.891 | (1) | 3.122 | 14.343 |
| Modality_Audible ~ Recipient visual attention_Yes | 3.527 | 0.500 | 2.522 | 0.012 | 1.323 | 9.402 |
| Modality_Silent-visual ~ Recipient visual attention_Yes | 4.754 | 0.494 | 3.155 | 0.002 | 1.803 | 12.532 |

n = 670 body acts.
"Modality_Audible" and "Modality_Silent-visual" were dummy coded and centred before entering random slopes in the model. The table shows odds ratios, standard errors, test results, P values and 95% confidence intervals. "(1)" Not indicated because of limited interpretation.

relative attraction among all possible bigrams was found between Rumble first combined with Ear-Flapping (pbin=10.103, $n = 33$, $P < 0.001$). The reversed order of Ear-Flapping first combined with Rumble occurred frequently ($n = 36$) but had a lower relative attraction (pbin=5.288, $P < 0.001$). The second highest relative attraction was found in Rumble first combined with Ears-Stiff (pbin=7.607, $n = 9$, $P < 0.001$). The reversed order of Ears-Stiff first combined with Rumble occurred frequently ($n = 22$) but had a lower relative attraction (pbin=1.953; $P < 0.05$). Significant relative attractions were also found for other combinations, mainly regarding body act types produced with the ears or tail (Table 3, Supplementary Data 4). A similar pattern of results was found in MDCA2, where the highest relative attraction was of Rumble and Ear-Flapping ($n = 34$; pbin=10.680, $P < 0.001$), followed by Rumble and Ears-Stiff ($n = 9$, pbin=10.313, $P < 0.001$). However, the inclusion of all overlapping signal cases resulted in the identification of other multicomponent combination types in MDCA2 (Table 3, See Supplementary Data 4).

## Do individual and social factors affect the order of combinations of vocalisations with gestures or body acts by elephants during greeting?

Overall, the fixed effects of the interaction between Signaller sex and Sex dyad and the average Nearest-neighbour index did not affect the order of vocalisations with gestures or body acts in multicomponent combinations ($\chi^2_2 = 2.839$, $P = 0.242$).

## Do individual and social factors affect the use of combinations of Rumble and Ear-Flapping by elephants during greeting?

Overall, the fixed effects of the interaction between Signaller sex and Sex dyad and the average Nearest-neighbour index affected the use of combinations of Rumble with Ear-Flapping ($\chi^2_2 = 6.034$, $P = 0.049$). Females used rumbles in combination with Ear-Flapping more frequently towards other females compared to males towards other males (Fig. 5, Table 4). However, the model explained a low proportion of variance (marginal $R^2 = 0.051$).

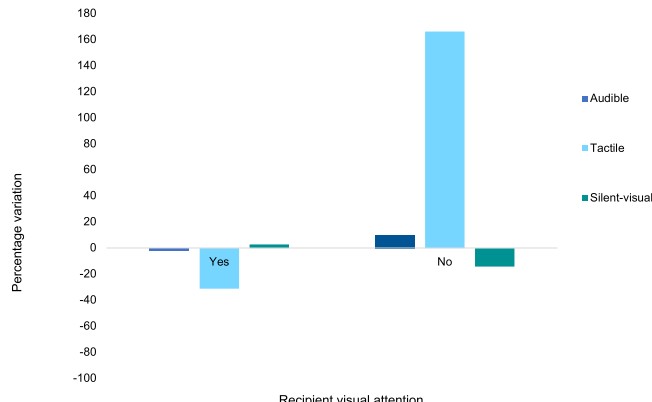

**Fig. 4 | Percentage variation in the use of body act modalities according to the recipient's state of visual attention during greeting.** Deviations above and below the 0 line show changes of modality according to the recipient's state of visual attention from the overall use of body acts. "Yes" indicates that the recipient was visually attending the body act; "No" indicates that the recipient was not visually attending the body act. $n = 670$ body acts (Silent-visual: $n = 398$; Audible: $n = 253$; Tactile: $n = 19$).

## Table 3 | Results of Multiple Distinctive Collocation Analysis MDCA1 and MDCA2

| Signal 1 | Signal 2 | Pbin Values MDCA1 | Pbin Values MDCA2 |
|---|---|---|---|
| Back-Towards | Rumble | 1.520 | 1.913 |
| Ear-Flapping | Rumble | 5.288 | 7.185 |
| Ear-Slight-Spread | Rumble | 2.284 | 2.848 |
| Ear-Spread | Rumble | 2.313 | 3.087 |
| Ears-Stiff | Rumble | 1.953 | 2.417 |
| Roar | Tail-on-Side | 1.696 | |
| Rumble | Back-Towards | 1.729 | 1.385 |
| Rumble | Ear-Flapping | 10.103 | 10.68 |
| Rumble | Ear-Slight-Spread | 1.442 | 1.744 |
| Rumble | Ear-Spread | 2.18 | 2.512 |
| Rumble | Ears-Stiff | 7.607 | 10.313 |
| Rumble | Head-Raise | 1.556 | |
| Rumble | Tail-on-Side | | 1.309 |
| Rumble | Tail-Stiff | 2.766 | 2.769 |
| Rumble | Tail-Waggling | 2.717 | 2.129 |
| Tail-on-Side | Roar | 1.476 | 1.62 |
| Tail-on-Side | Rumble | | 1.458 |
| Tail-Stiff | Rumble | | 1.542 |
| Tail-Touch | Trumpet | | 1.314 |
| Tail-Waggling | Rumble | 1.537 | 2.061 |
| Trumpet | Ear-Slight-Spread | 1.844 | 1.495 |
| Trumpet | Tail-Raise | 1.954 | 1.860 |

The table shows significant Pbin Values of the bigrams of vocalisation and body act types produced by the subjects during greeting (Sample sizes: $n_{MDCA1} = 337$; $n_{MDCA2} = 403$).
Interpetation of pbin values: pbin_*>3 = > $P < 0.001$; pbin_*>2 = > $P < 0.01$; pbin_*>1.30103 = > $P < 0.05$.

## Discussion

We present a comprehensive study of elephant multimodal communication during greetings, with evidence that elephants target gestures at their audience selecting their modality appropriately according to the audience's state of visual attention. We demonstrate that elephants greet with specific vocalisations and gestures or body acts of different modalities, which they integrate into multicomponent combinations. Additionally, we present evidence that individual and social factors shape the use of rumble vocalisations and ear-flapping gestures in combination.

When elephants reunited, they greeted with rumble, roar, and trumpet vocalisations, with gestures like Ear-Flapping, Ear-Spread, Ears-Stiff, Back-Towards, Trunk-Reach, and with body acts like Tail-Raise and Tail-Waggling. Furthermore, elephants often exhibited temporal gland secretions and/or urinated. Our results are consistent with previous descriptions of the greeting behaviour of related or closely-bonded wild female elephants and of related female zoo elephants meeting after years apart[30,63]. However, the greetings of our male elephants differed from those used in the wild, where males typically only direct their trunks to scent-emitting organs and/or may rumbe[30,55]. Our males greeted both each other and females using the same elaborate greetings as close bonded female elephants[30].

The functions of greetings are varied. In spotted hyenas, chimpanzees, or bonobos, greetings can signal dominance status, while in wild dogs and capuchin monkeys they may help promote group cohesion or coordination[56,77]. While the proposed function of the elaborate greetings of closely bonded female elephants is to promote recognition and strengthen social bonds[20,30], a recent study suggested that male elephants direct their trunk to other males to facilitate positive interactions or assess chemical information upon reunion[55]. However, that study had no information on the males' social bonds, but our subjects live in a tighter social group than males in the wild[47]. Our results suggests that social relationships flexibly impact the use of signals by elephants during greeting, and supports the hypothesis that elaborate greeting behaviour functions to strengthen social bonds upon reunion, including among closely bonded semi-captive males[30].

First-order intentionality is a fundamental property of human language that allows us to express meaning and is an essential precursor to second-order intentionality[31,33,34]. Today we know that all non-human apes use large gestural repertoires with first-order intentionality[39]. However, evidence in other animals, including non-anthropoid primates[78–80], is scarce and/or restricted to a few signals. For example, coral reef fishes use a referential gesture to indicate prey during cooperative hunting, while Arabian babblers use object presentation and babbler walk for joint travel[81,82]. A

first step in identifying first-order intentional use is determining whether signals are directed at a specific audience[36,83]. We found that elephants targeted most body act types towards conspecific recipients after visually checking them, and used silent-visual and audible body act types when recipients were visually attending. Moreover, when recipients were visually attending, signallers were more likely to choose a silent-visual body act as compared to when they were not attending. In contrast, when recipients were not visually attending, signallers preferentially selected a tactile or an audible body act as compared to when they were attending. Wild chimpanzees show similar adjustment, selecting silent-visual gestures more often when recipients are visually attending and tactile gestures when they are not. In contrast, no adjustment is observed in their use of audible gestures, presumably because recipients can acquire audible information whether they are visually attending or not[39,58]. Our results therefore provide evidence that most body act types produced during greeting represent audience-directed gestures, supporting the presence of first-order intentionality in elephant gestural communication.

Some tail body act types that could be defined a priori only as silent-visual were produced regardless of whether the recipient would be able to perceive them and were, therefore, not considered as gestures by our current definition. At present, the alternative parsimonious interpretation is that these tail actions are non-directed cues or signs of emotional arousal. However, these tail actions were sometimes accompanied by urination or defecation, and some of them were most frequently produced when the recipient was a few metres away. These results raise the possibility that some tail actions in elephants may serve an alternative function as possible olfactory gestures by sending, emphasising, or inviting the recipient to access scent-based information in the genital area[19].

**Fig. 5 | Probability of the combined use of Rumble and Ear-Flapping according to signaller sex and whether he/she was greeting a recipient of the same or opposite sex.** The bars represent the probability of combination. The horizontal lines with error bars depict the fitted model lines and their bootstrapped confidence intervals for each combination of Signaller sex and Sex dyad. $n = 337$.

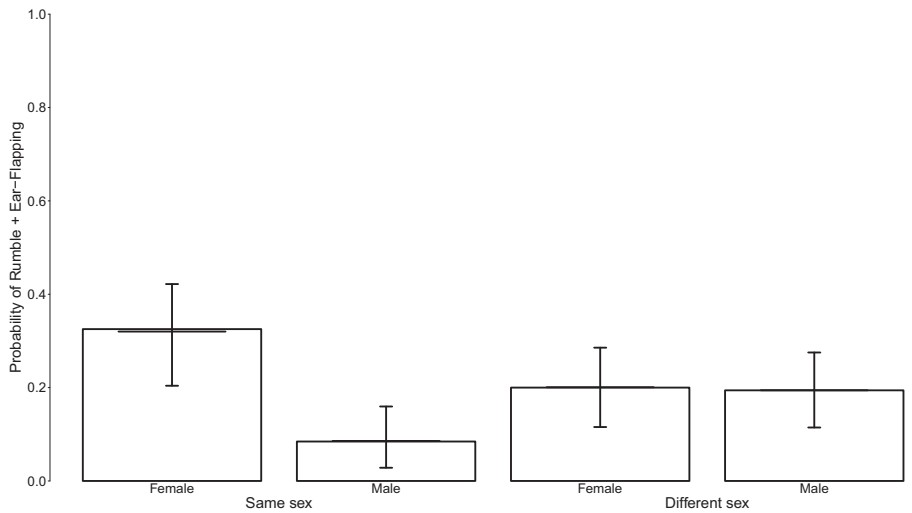

**Table 4 | Results of GLMM2 exploring the effect of the interaction between Signaller sex and Sex dyad and the average Nearest-neighbour index on the use of Rumble with Ear-Flapping by the subjects during greeting**

|  | estimate | SE | Lwr CI | Upr CI | χ² | P | min | max |
|---|---|---|---|---|---|---|---|---|
| Intercept | −0.754 | 0.239 | −1.308 | −0.325 |  | (1) | −0.923 | −0.504 |
| Signaller sex_Male | −1.613 | 0.492 | −2.929 | −0.767 |  | (1) | −1.934 | −1.088 |
| Sex dyad_Different | −0.628 | 0.368 | −1.419 | 0.103 |  | (1) | −0.916 | −0.087 |
| Nearest-Neighbour index | −0.079 | 0.126 | −0.339 | 0.179 | 0.397 | 0.529 | −0.2 | 0.057 |
| Signaller sex_Male:Sex dyad_Different | 1.572 | 0.619 | 0.49 | 3.067 | 5.614 | 0.018 | 0.842 | 1.862 |

$n = 337$ multicomponent combinations.

"Nearest-Neighbour index" was z-transformed before entering the model, whereas "Signaller sex" and "Sex dyad" were dummy coded and centred before entering random slopes in the model. The table shows estimates, standard errors, bootstrapped confidence intervals, test results, and minimum and maximum of the model stability estimates after removing levels of random effects one at a time. Significant results are highlighted in bold. "(1)" Not indicated because of limited interpretation.

We found that elephants combined different vocalisations with gestures in different ways and orders. The most frequent combination was of Rumble vocalisations with Ear-Flapping gestures, with Rumble most often first and Ear-Flapping second. The second most frequent combination was of rumbles with Ears-Stiff gestures. However, the physical forms of Ears-Stiff and Ear-Slight-Spread are considered indications of listening behaviour in elephants[20]. Thus, elephants may slightly open their ears when vocalising to facilitate hearing a potential response rather than for communication.

Information-bearing combinations with syntactical properties have been identified in a few animal vocal sequences[84]. Some birds or monkeys combine functionally distinct vocalisations into compositional combinations whose function is related to the functions of the parts[85,86]. A recent study has suggested that the scarce evidence for compositionality in animal communication might be due to its exploration at the unicomponent level[87]. However, evidence that apes combine vocalisations with non-vocal signals to elicit different reactions in recipients and, thus, convey different meanings (or goals) is scarce[17,88]. Elephants combine different vocalisations in different orders, a pre-requisite of syntactic abilities, but whether these orders convey syntactical meaning remains unknown[89]. By showing that elephants combine vocalisations and gestures in specific ways and orders, our study represents a first step towards exploring syntactic properties in elephant multicomponent combinations.

Contrary to our prediction that vocalisations may serve as attention getters to subsequently incorporated gestures[15], we found no order-effect within multicomponent combinations. Moreover, because greeting does not involve a request for a change in behaviour of the recipient, we were unable to determine whether specific multicomponent combinations and orders elicited specific reactions in recipients. We suggest, however, that

Rumble and Ear-Flapping may be combined in a redundant way. Ear-Flapping is a single multisensory gesture conveying visual and audible information, as well as possible olfactory information from the temporal glands (e.g., individual identity, reproductive state, or arousal state) via scent wafting[19,90]. Rumbles contain information on individual identity[22,23], sex[24], age[25], reproductive state[26,27], and emotional state[28]. Thus, their combination may provide redundant multisensory information about the signaller salient to recipients upon reunion.

Much research on multimodal communication has focused on its function for reproductive purposes[8]. In particular, pair-bonded birds or primates use multisensory combinations to advertise and reinforce pair bonds[91,92]. We found that Ear-Flapping and Rumble were most frequently combined during female greetings, confirming previous descriptions in related and closely bonded wild female elephants[20,30]. In our semi-captive group, three out of four females' strongest association partner was another female, mirroring natural social ties[46,48]. In addition, our elephants are all under contraception to avoid births in semi-captivity. Thus, the combination of Ear-Flapping and Rumble, and possibly of other gestures and vocalisations, during greeting seems more likely to serve enhanced recognition and bonding of socially bonded elephants upon reunion, rather than any reproductive purposes.

Our study shows that elephant greetings are a constellation of vocalisations, audience-directed gestures, and multicomponent combinations conveying information to various sensory channels that may serve to promote individual recognition and social bonding. Elephants are physically distinct and distantly related to our ape family, but they share with us a multi-level social system, a long lifespan, and sophisticated cognition[93]. Finding audience directedness, a core property of first-order intentional

**Article**

communication, in a range of elephant gestures, and specific multi-component combinations of vocalisations and gestures as shown in other primates[14,39], suggests convergent evolution of these capacities across distant species with similar cognitive and social niches. Future studies should explore the impact of social relationships on signal use in wild male and female elephants, the meanings of elephant gestures, and the use of multi-component combinations in contexts that involve explicit changes in recipient behaviour. The impact of vocalisations and gestures in isolation, in combination, and in different orders on recipient behaviour could be explored to understand whether multicomponent combinations provide redundancy, communicative flexibility, or specific combinatorial meanings. Lastly, future research should explore the effects of multicomponent and multisensory combinations on recipients to help elucidate the functions of the production and perception of multimodal signalling in elephants.

## Data availability

The datasets used to conduct statistical analyses are available on Github[94]. Source data underlying Figs. 3, 5, and Supplementary Figs. 1, 2, 4 can be found on Github[94]. Source data for Fig. 4 and Supplementary Fig. 3 can be found in Supplementary Data 2 and Supplementary Data 3.

## Code availability

The R code used to conduct statistical analyses is available on Github[94]. Statistical analyses were performed with R software version 4.0.2 and the following packages: *lme4* version 1.1–23[95,96]; *MuMIn* version 1.43.17[97]; *mclogit* version 0.9.6[98], *stats* version 4.0.2[96]. Collocation analyses were performed using Coll.analysis V 3.2a scripts provided by Gries[70].

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

## Acknowledgements

The authors thank the elephant carers and all the staff at elephant CREW for their assistance in the field. We thank Katharina Prager and Yvonne Nyaradzo Masarira for their assistance in data collection. We thank Megan Pacifici for her scientific illustrations of elephant body acts. This research was funded by the Austrian Science Fund (FWF) [AW0126211] and the European Union's 8th Framework Programme, Horizon 2020 [802719]. For open access purposes, the author has applied a CC BY public copyright license to any author-accepted manuscript version arising from this submission.

## Author contributions

Vesta Eleuteri: Conceptualisation, Investigation, Methodology, Formal analysis, Writing - Original Draft; Lucy Bates: Methodology, Writing—Review & Editing; Jake Rendle-Worthington: Resources, Writing—Review & Editing; Catherine Hobaiter: Conceptualisation, Methodology, Supervision, Funding acquisition, Writing—Review & Editing; Angela Stöger: Conceptualisation, Methodology, Supervision, Funding acquisition, Writing—Review & Editing.

## Competing interests

The authors declare no competing interests.
