## [Peer review file · Communications Biology]

Reviewers' comments:

Reviewer #1 (Remarks to the Author):

Review Vesta et al.

This paper describes the production of signals and signal combinations in captive African elephants. The data were collected after brief separations of dyads of elephants that occurred repeatedly over two months. The analyses are based on 1282 instances of signaling (considering only signals that occurred at least twice) during a total of 89 "greeting communications." The authors found that some signals occur together, such as "rumble" and "ear flap," evidencing multimodal communication. They also report that the communication of elephants fulfills the criteria for first-order intentionality.

Although the topic of multimodal communication appears to attract continued attention, the field is long beyond simply reporting that it exists. It has long been known that signals in different modalities may co-occur or that some signal production generates signals in different modalities (e.g., stridulation is visible and audible). More exciting questions would be how recipients integrate information from different sensory channels or the functional significance of such multimodal communication. The paper does not speak to these conceptually more exciting issues. Despite some limited hypothesis testing, it is mainly descriptive. Descriptive studies can be precious, but because of the experimental approach in eliciting the communicative behavior, and the fact that the animals live in captivity, the value for understanding the animals' natural history is limited. The manuscript is also excessively long, and the discussion focuses almost exclusively on elephant communication. Thus, the paper is not of broad appeal.

Diagnosing "first-order intentionality" is also not very surprising. What would be the alternative? It must be expected that signalers pay attention to whether their signals have the intended effects, and then it is a question of how they do so. Are these evolved behaviors, or have the animals learned that visual signals do not elicit responses when the recipient is not looking?

Thus, it is neither surprising nor conceptually intriguing to find that some of the signals co-occur or that the recipient's attentional state modulates signal use. Therefore, I would suggest submitting the paper to a more specialized journal. Before they do so, the authors should strip down the paper to the key messages and provide a better rationale for their study. Simply saying, "The functions of multimodal communication, however, remain unclear" (line 33) is firstly not correct and secondly not a good motivation when the paper does not address the function. The authors themselves note that "Despite the divergence in definitions, the core of all multimodal communication is that signal modalities are integrated for communicative purposes, to back-up, enhance, or refine the information transmitted in single modalities^{4,13}." (line 87)

The authors then argue, "However, a systematic study of multimodal signaling has never been conducted in elephants." Again, that's a poor justification. It would be better to motivate the study theoretically/conceptually. Many things have not been studied systematically, but the reader would like to know what they can learn about the principles of animal communication, where the points of contention are, and how a study addresses these issues.

Some issues with the terminology and argumentation also need to be fixed. For instance, the authors write, "Animals communicate using different modalities of sensory information [...] that are encoded

into a wide range of signal types". Modalities are not encoded.

The likening of the combination of different signals to human sarcasm is off. Also, there is a difference between disambiguation via signal combination or contextual modification.

The writing is sometimes hard to follow; thus, I am unsure whether there are circularity issues. Specifically, the authors write, "We then defined socially directed "gestures" as those mechanically ineffective body acts that met the first-order intentional criterium (sic!) of audience directedness" – but if they define gestures that way, they must – by definition – be first-order intentional. Therefore, the conclusion is set up by the definition at the outset.

Line 68: Unfortunately, the authors repeat the misconception of multimodal signaling that has crept into the ape gesture literature, where facial and gestural communication is conceived as different "signaling channels." They are not.

Line 106: how does the present study contribute to a better understanding of the evolution of multimodal communication?

135: "linguistic property of first-order intentional use in gestures" – first-order intentional use, according to the criteria reported in this paper, is not exclusively a linguistic property. One must always consider more straightforward alternative explanations for the observed use of signals.

150: Why is the question of audience-directedness interesting?

186: "they" could also refer to the elephants, but the authors mean the signals.

196: it would be very surprising if the signals were combined randomly. The authors are knocking down a straw man.

Table 1: the numbers are difficult to make sense of when one does not know the entire range of values.

290: Synchronize (typo)

306: Criterion

309: What are signs of the signaler "visually checking the recipient"? A clear definition of the behavior needs to be given.

318: The authors should keep the treatment of the beginning and end of sequences separate from the discussion of first-order intentionality.

338: What are "tail body act types"?

352: "The communication level" is an odd category.

388: How was the criterion of "at least twice" established?

405: Binomial test cannot be used on non-independent data (pseudoreplication). The entire paragraph raises questions about whether the definitions applied predetermine the outcome. The analyses suggest that that was not the case, but the authors should be clear about this.

417: Chi-squared tests cannot be used on non-independent data. See above.

452: The purpose of the two different collocation analyses remains unclear.

497: The question of the order of signals was not set up in the introduction. Why would this be interesting? What does “fairly stable” mean? Please give numbers.

Before submitting elsewhere, I would suggest moving the ethogram and the cumulative distribution of signal types to the ESM.

Table 3. “communications” is an odd category, especially for “Urinating”.

564: The authors make a big deal that 11/19 tactile signals were used when the recipient was visually attending, in contrast to 89% of “silent visual body acts.” Yet, the authors also found that 83% of audible body contacts were used when the recipient was visually attending, although one would not predict this to be necessary. This latter result speaks against the author’s hypothesis.

Figure 3 fails to acknowledge the small sample size for tactile signals. Moreover, the y-axis label is missing. Overall, this way of presenting the data in % deviation seems to be exaggerating the differences.

587: What are “tail body acts”?

What is the purpose of the analysis presented in Figure 5?

As mentioned above, the discussion is restricted to elephant communication, and the broader significance of the findings remains elusive.

802: delete the second “the.”

Reviewer #2 (Remarks to the Author):

This paper seeks to increase our understanding of elephant communication by specifically investigating the use of multimodal signals in a semi-captive group during reunion events. Additionally, this paper seeks to add to the growing literature investigating the form, function, and evolution of multimodal communication more generally.

Overall, the paper is very thorough and well-written. The authors provide detailed descriptions of multimodal signals within the elephant communicatory repertoire, which bear recognition and further research. I have only a few suggestions-

Title-

(1) I think "semi-captive" should be added to the title of the work, to indicate this work is done with a non-wild population.

Abstract-

(2) If possible, it would be nice to give a few details about the methods here (1-2 sentences, mentioning that these captured reunion episodes, and the types of observations/models that were used).

Introduction -

(3) It might be helpful for the Introduction to be separated into sub-sections. There is a lot of really important information in here, covering a wide-range of topics. It would help orient the reader to have this information broken down into labeled sub-sections based on primary topics.

Methods-

(4) Study site and subjects- This sub-section might read better if you switched the order of the current first and second paragraph, such that lines 224-234 come first in this sub-section, followed by lines 215-223.

Discussion-

(5) It is unclear how this paper adds to our understanding of the evolution of multimodal communication (line 791-792, and also mentioned in the abstract) - perhaps the authors should add a sub-section to the discussion that covers this perspective more explicitly, or if the topic is not explicitly covered in this paper then perhaps this wording should be adjusted.

(6) The authors seem to suggest that multimodal signals used by elephants during reunions act to strengthen bonds between individuals (lines 777-790). There are a number of papers about multimodal signal use for pair-bond maintenance and/or advertisement in the literature (including in various monogamous fish, birds, and primates; for example, see Singletary & Tecot (2020) "Multimodal pair-bond maintenance: A review of signaling across modalities in pair-bonded nonhuman primates"). While this area of research focuses largely on bonds between male-female mating pairs, it may be worth mentioning some of this work, and potentially making recommendations for future research that investigates this particular possible function (i.e., bond promotion for social rather than reproductive functions) in elephants.

(7) It might be helpful to add a "Future Studies" sub-section to have some more explicit suggestions for where/how future research can be conducted to answer the remaining questions your paper introduces.

(8) It seems like there are a number of potential limitations to this study which are not discussed - this should be added as a "Limitations" sub-section.

Reviewer #3 (Remarks to the Author):

The manuscript entitled "Multimodal communication in the greeting behaviour of African savannah elephants" describes a most interesting study on multimodal communication (greeting behaviour) in a population of semi-captive elephants. The authors collected an impressive amount of behaviours, and found evidence of first-order intentionality in elephant gestures.

The study is timely, the manuscript is well written and comprehensive, and it can be of interest for a wide audience. As such, I mostly have requests for clarifications or suggestions on how to improve the manuscript readability, together with a few more substantial requests. The latter concerns details about the methodology, and especially how the data were coded.

Major points:

- In the introduction, the discussion on the greeting behaviour of wild elephants refers either the elephants separating and reuniting regularly (as a result of living in fission-fusion societies) or after periods of separation. While the experiment is based on separation and reunion procedure, I am wondering how comparable these situations are with the wild animals. More precisely: the separation in this experiment was rather short; do we expect differences in the greeting behaviour compared to much longer separations (eg, days?)
- I missed in the methods several details on how the data were coded, especially considering that coding animal behaviour from videos can be a rather subjective endeavour:
 - a) was there only one rater, or more than one?
 - b) If more than one, did the authors calculate inter-rater reliability or similar measures?
 - c) When coding greeting in a pair, was each animal considered simultaneously as a possible signaller and recipient?
- Still in the methods: It is not clear to me how the distance during greeting was measured. Please provide more information (e.g., were there visible markers that the coders could use?)

Minor points:

- Lines 61 and following: the authors use sarcasm as an example of communication requiring both linguistic as well as paralinguistic cues to be deciphered. While I very much agree on this, I am not convinced that this is the best example for multi-modality: one can be sarcastic in writing, or even in face-to-face communication without adding gestures. I would suggest using a clearer example. I would also distinguish being sarcastic from making a joke, as these two terms can indicate rather different concepts.
- Line 238: Could the authors add a few lines/give a few details in the manuscript to explain what the STRANGE framework is?
- Lines 264 and following: While I think it makes intuitive sense to selected animals showing strongest social bonds, I suggest to add a few lines to explain the rationale for this choice as, from the introduction, it wasn't clear to me that greeting behaviours were occurring more in socially closer animals.
- Line 547: "in 6% it was unknown whether elephants produced them" – I am not sure I understand the sentence. Please clarify.
- Figure 3 and 4: Please reposition the labels "yes" and "no" on the x axis so that they are not inside the bars
- Line 612: The acronym MDCA was already explained, there is no need to repeat it here.
- Line 644: "Nearest-neighbour index affected the use of combinations of Rumble with Ear-flapping ($\chi^2=6.034$, $P=0.049$)."
- Discussion, lines 793-795: The conclusion is rather vague, and could be improved by adding some examples of "promising venues" for future research or implications.

Response to Reviewers

Reviewer #1 (Remarks to the Author):

Review Vesta et al.

1.

This paper describes the production of signals and signal combinations in captive African elephants. The data were collected after brief separations of dyads of elephants that occurred repeatedly over two months. The analyses are based on 1282 instances of signaling (considering only signals that occurred at least twice) during a total of 89 "greeting communications." The authors found that some signals occur together, such as "rumble" and "ear flap," evidencing multimodal communication. They also report that the communication of elephants fulfills the criteria for first-order intentionality.

Thank you very much for your feedback and for your very constructive comments on how to improve our paper.

2.

Although the topic of multimodal communication appears to attract continued attention, the field is long beyond simply reporting that it exists. It has long been known that signals in different modalities may co-occur or that some signal production generates signals in different modalities (e.g., stridulation is visible and audible). More exciting questions would be how recipients integrate information from different sensory channels or the functional significance of such multimodal communication. The paper does not speak to these conceptually more exciting issues.

Thank you for your comment. We agree that these questions are exciting. Understanding signal modality integration typically requires use of playback experiments in which modalities are manipulated, while understanding function (e.g., redundancy or refinement) requires exploring the use and effects on recipients of signals when used in isolation and when combined together (Partnan & Marler, 1999, 2005). We believe that starting with a description of multimodal or multicomponent signal use is an important first step to understanding the functions of multimodal signals or multicomponent combinations in elephants. We could not directly explore the questions you propose here as the context of production (greeting) does not involve the request of a specific reaction in the recipient, which is necessary to explore the effects of components in isolation and in combination on recipient behaviour. But to address this broader point, we now expand on the functions of multimodal communication in animals in the introduction (Lines 65-97) and address in the discussion some possible functions of the vocal-gestural combinations we found during elephant greetings (Lines 708-729). We also mention the limitations of our study and suggest that future studies explore the proposed questions to enhance our understanding of the functions of multicomponent combinations in elephant communication (See lines 699-701; 728-729; 744-752).

3.

Despite some limited hypothesis testing, it is mainly descriptive. Descriptive studies can be precious, but because of the experimental approach in eliciting the communicative behavior, and the fact that the animals live in captivity, the value for understanding the animals' natural history is limited.

Thank you for raising this issue. We recognize that studies on wild elephants are more ecologically valid to understand their natural history. However, like many semi-captive elephants, our subjects come from the wild and roam in an open environment where they encounter wild elephants on a daily basis, so they are exposed to wild elephant behaviour and interactions, highlighting the higher ecological validity of our study (as compared to studies on e.g., zoo

elephants). Moreover, separation-reunion procedures are a method used to stimulate greeting behaviour in elephants and facilitate detection of vocalisations (which is extremely challenging during the elaborate greetings in the wild), but are ethically possible and feasible only with captive or semi-captive elephants.

We found that our semi-captive elephants greeted in a similar way to what has been previously described in wild female elephants, as well as zoo female elephants reuniting after years of separation (See Lines 595-599). Together with our results, this highlights that greeting displays are most-likely a species-specific behaviour in elephants, with similar forms retained in wild and close-bonded captive elephants. We believe these findings contribute to our understanding of their natural history and have important implications for elephant welfare in captivity.

4.

The manuscript is also excessively long, and the discussion focuses almost exclusively on elephant communication. Thus, the paper is not of broad appeal.

Thank you for your suggestion. We reduced the length of the manuscript and hope that we have now broadened the appeal by incorporating a wider range of studies on other species as well as on human language in the introduction and discussion (See lines 59-97, 122-162, 619-624, 639-643, 652-656, 684-693).

5.

*Diagnosing “**first-order intentionality**” is also not very surprising. What would be the alternative? It must be expected that signalers pay attention to whether their signals have the intended effects, and then it is a question of how they do so.*

Thank you for raising this issue. On the one hand we fully agree. However, as researchers who have spent a substantial proportion of our work building the case for intentionality in non-human species' communication, we are also familiar with the fact that this continues to be considered a human-specific capacity (e.g., Scott-Philips & Heinz, 2023 PNAS). Moreover, decades of research efforts on – for example – vocal communication in other species, failed to find evidence for intentional use (Rendall et al., 2009; Seyfarth & Cheney, 2017). We have expanded on our discussion of intentionality to highlight the importance of our findings in the introduction and discussion. While first-order intentionality (i.e., goal-directed intentionality) has been diagnosed in large repertoires of ape and some other primate gesture, only a couple of studies have explicitly shown evidence for this ability in non-primate species, and only in a few individual signals. The alternative to intentional production is that body acts are fixed responses to stimuli or expressions of emotional states. We believe that elephant communication is a particularly promising system to explore first-order intentional communication, because elephants – like apes – live long lives in a richly structured society in which there are strong differentiated bonds amongst both kin and non-kin, and in which flexible communication might be needed to meet their different social goals. In addition, observational descriptions of the many body acts used by elephants across different behavioural and social contexts provide strong potential for use as intentional gestures (i.e., body acts showing first-order intentionality). However, no study has explicitly tested whether the body acts described in elephants are actually intentional gestures, targeted at recipients, nor whether they are used to achieve different goals. We believe that to understand the communicative function of these acts it is important to first address if these are targeted at recipients. We add more context on the importance of studying and finding intentionality in elephants in lines (Lines 122-162; 225-229; 638-660; 731-743).

6.

Are these evolved behaviors, or have the animals learned that visual signals do not elicit responses when the recipient is not looking?

Intentionality is an evolved capacity that appears in human and ape development around 9-12 months of age. We suspect it is the same for elephants, but future studies are needed to explore the ontogeny of intentional communication across individuals and contexts in elephants to properly establish this. We have added this information in Lines 132-133.

7.

Thus, it is neither surprising nor conceptually intriguing to find that some of the signals co-occur or that the recipient's attentional state modulates signal use.

We now provide more context on why it is intriguing to explore and find this capacity in a species so far away from our primate lineage (Lines 143-162; 731-743).

8.

Therefore, I would suggest submitting the paper to a more specialized journal. Before they do so, the authors should strip down the paper to the key messages and provide a better rationale for their study.

Thank you for this suggestion. We have stripped down the paper to the key messages and now provide a better rationale for studying intentionality and multicomponent combinations in elephants (See Introduction).

9.

*Simply saying, "The **functions of multimodal communication**, however, remain unclear" (line 33) is firstly not correct and secondly not a good motivation when the paper does not address the function. The authors themselves note that "Despite the divergence in definitions, the core of all multimodal communication is that signal modalities are integrated for communicative purposes, to back-up, enhance, or refine the information transmitted in single modalities^{4,13}." (line 87)*

Thank you for noticing this inconsistency. We removed the sentence and edited the abstract to avoid confusion (Line 33). We now expand on the functions of multimodal communication in the introduction and discussion and have added a section on the possible functions of our multimodal greetings in the discussion (See lines 73-97; 708-729).

10.

The authors then argue, "However, a systematic study of multimodal signaling has never been conducted in elephants." Again, that's a poor justification. It would be better to motivate the study theoretically/conceptually. Many things have not been studied systematically, but the reader would like to know what they can learn about the principles of animal communication, where the points of contention are, and how a study addresses these issues.

Thank you for this important suggestion. We now situate our study more theoretically by adding sections on the functions of multimodal communication in animals and the importance of intentionality in human language and ape communication and that making a case for this in elephants suggests convergent evolution of this important capacity in cognitively and socially complex species (See edited Introduction please).

11.

Some issues with the terminology and argumentation also need to be fixed. For instance, the authors write, "Animals communicate using different modalities of sensory information [...] that are encoded into a wide range of signal types". Modalities are not encoded.

Line 55-56: We corrected the sentence. We also clarified the terminology of the study by editing it throughout and adding Table 1 with all the key terms of the study.

12.

The likening of the combination of different signals to human sarcasm is off. Also, there is a difference between disambiguation via signal combination or contextual modification.

Lines 59-63: We removed the example about sarcasm and made a more general point on how humans modify the content of speech by combining it with facial expressions or gestures.

13.

The writing is sometimes hard to follow; thus, I am unsure whether there are circularity issues. Specifically, the authors write, “We then defined socially directed “gestures” as those mechanically ineffective body acts that met the first-order intentional criterium (sic!) of audience directedness” – but if they define gestures that way, they must – by definition – be first-order intentional. Therefore, the conclusion is set up by the definition at the outset.

Thank you for highlighting our lack of clarity here. Gestures are unusual in that, in addition to their form, they are defined by their intentional production (See Byrne et al., 2017; Rodrigues et al., 2021). Thus, we use the term body acts to refer to bodily signals (as compared to vocalisations, or chemical signals) and, within these, we discriminate “audience directed gestures” as being those body acts that are produced intentionally by being targeted at a recipient. We have now edited the sentence (Lines 288-295) and changed the terminology throughout to “audience directed gestures” (See Table 1) for clarity.

Note that the body acts that did not meet the criterion of audience directedness were not considered gestures, see also Table 2 under “Signal”). We now clarified the section “Definitions of signal types and modalities” and added Table 1 with definitions of the core terms of the study.

14.

Line 68: Unfortunately, the authors repeat the misconception of multimodal signaling that has crept into the ape gesture literature, where facial and gestural communication is conceived as different “signaling channels.” They are not.

We acknowledge the confusion and have now clarified our use to describe vocalisations, gestures and facial expressions as signals produced with different articulators (body parts) (Line 67).

Moreover, in the introduction we clarify the difference between multimodal communication, multisensory combinations and multicomponent combinations following previous studies to avoid confusion (Lines 65-70). We also added all these definitions in Table 1.

15.

Line 106: how does the present study contribute to a better understanding of the evolution of multimodal communication?

Thank you for raising this issue. We added how our study contributes to the evolution of multimodal communication in the introduction and conclusion of the discussion. See Lines 45-49; 225-229; 732-743.

16.

135: “linguistic property of first-order intentional use in gestures” – first-order intentional use, according to the criteria reported in this paper, is not exclusively a linguistic property. One must always consider more straightforward alternative explanations for the observed use of signals.

Thank you for this suggestion. Typically, evidence of intentional use in communication requires coding of many cases of the mechanically ineffective body acts and exploration of the intentionality criteria. If a body act is mechanically effective and does not meet one or more of the criteria it is not considered a gesture. We consider alternatives - both for body acts that meet this criterion (e.g., Ear-Stiff) and for body acts that do not (tail body acts). See lines 604-610; 661-674, 680-683.

17.

150: *Why is the question of audience-directedness interesting?*

Thank you for the question. Audience directedness is typically the first criterion used to establish intentional use in animal signals, and allows us to discriminate whether a body act is a communicative gesture (used intentionally) or a body signal (used reflexively or without an intention to convey information to or modify recipient behaviour). To explore whether the large repertoire of body acts used during greeting are gestures, we need to first assess whether they are directed at the audience. To explain this in more detail we have expanded the section on intentionality in lines 122-163) in the introduction, where we add information on intentionality and audience directedness.

18.

186: *“they” could also refer to the elephants, but the authors mean the signals.*

Edited now (Lines 195-197).

19.

196: *it would be very surprising if the signals were combined randomly. The authors are knocking down a straw man.*

The MDCA analysis allows us to explore whether signals are used in random combinations or not (“randomly” is the terminology used in this analysis rather than our own). We do agree that straightforward signal pairs will frequently show an order (e.g., a whimper followed by a reach to ask for an object in chimpanzees). But greetings in elephants are described as a chaotic plethora of signals (please see video of elephant greeting in Supplementary Information) and the most parsimonious prediction is that they are combined randomly in this context. Based on the large number of types and tokens produced in elephant greetings, we did not predict that elephants would combine signals in specific ways or orders in this context, but our results show otherwise.

Exploring whether signals are ordered randomly is a first step used to explore syntactical properties in animal communication, and was done so to show similar properties in elephant vocal communication (Pardo et al., 2019). Therefore, we believe it is of interest that we found non-random use and order of signals in elephants. We added further context on this in lines 676-701.

20.

Table 1: *the numbers are difficult to make sense of when one does not know the entire range of values.*

We removed the table to shorten the manuscript but put it in the Supplementary Information along with a Supplementary Calculations 1 the calculations with the entire range of values.

21.

290: *Synchronize (typo)*

Line 272: Now corrected.

22.

306: *Criterion*

Thank you for noticing the mistake. We now corrected it throughout.

23.

309: *What are signs of the signaller “visually checking the recipient”? A clear definition of the behavior needs to be given.*

Thank you for the suggestion. We were referring to the analyses (binomial tests) exploring whether signallers use the body act types when visually attending the recipient. We now clarified this referring to the statistical analyses (Lines 292-295). Please see lines 334-341 for more details on our description of signaller and recipient visual attention.

24.

318: *The authors should keep the treatment of the beginning and end of sequences separate from the discussion of first-order intentionality.*

We moved this section to the Supplementary Information to shorten the manuscript (“Additional information on Definitions of signal types and modalities”). See edited MS section “Definitions of signal types and modalities” (Line 275-295).

25.

338: *What are “tail body act types”?*

Line 501: Now changed to “body acts produced with the tail (i.e., tail body acts)” to clarify.

26.

352: *“The communication level” is an odd category.*

Line 322: Now edited.

27.

388: *How was the criterion of “at least twice” established?*

We followed studies of gestural repertoires in ape species; e.g., Hobaiter & Byrne 2011. We have added more information and the reference in lines 357-359.

28.

405: *Binomial test cannot be used on non-independent data (pseudoreplication). The entire paragraph raises questions about whether the definitions applied predetermine the outcome. The analyses suggest that that was not the case, but the authors should be clear about this.*

Thank you for raising this issue. We realized we violated the assumption of independence as signals were produced by the same signallers. However, due to the small sample sizes per body act type, we could not conduct an analysis controlling for Signaller.

The issue of pseudoreplication in small sample studies is one that has been raised in the field (e.g., Waller et al., 2013 *Animal Behaviour*) and one we are sensitive to – and have addressed in other work (e.g., Hobaiter & Byrne, 2011). As was recently pointed out (Whitehouse et al., 2023 *IJP*) the field has largely addressed this through the use of multi-level models. However, these require larger datasets that are not currently available. As a result, we used binomial models as we felt that – while not ideal – these were the only method available to test our data. We highlight this violation now in the methods (lines 382-385).

29.

417: *Chi-squared tests cannot be used on non-independent data. See above.*

Again, thank you for raising this issue. Here we had sufficient sample size to conduct a GLMM with binomial error structure and logit link function to investigate whether elephants choose their body act modality appropriately according to the recipient’s state of visual attention. In doing so, we deal with non-independence by including Signaller as random effect (See lines 386-397 in Methods and 523-530 in Results).

30.

452: *The purpose of the two different collocation analyses remains unclear.*

To capture both the frequency of vocalisations and gestures elephants combine and the order in which they combine them we conducted two separate MDCAs.

- In the first MDCA, we explore the specific order in which vocalisations and body acts are combined. Here if, for example a vocalisation overlapped with more than one body act

produced with the same body part (e.g., Rumble overlapping first with Ear-Spread and then with Ear-Flapping), we kept only the first overlapping body act to the vocalisation. We do so because the start time of the following body act (here Ear-Flapping) would necessarily depend on the end time of the previous one (here Ear-Spread) and it would, thus, necessarily follow the vocalisation, impacting our ability to explore order effects in vocal-gestural combinations.

- In the second MDCA, we kept all combinations of vocalisations and body acts to understand the overall frequency of co-occurrence of vocalisations and body acts (without considering any pattern of ordering between the signals).

We have edited lines 425-436 to make this clearer.

31.

497: The question of the order of signals was not set up in the introduction. Why would this be interesting?

We were interested in exploring whether vocalisations were produced before gestures in the combinations and, thus, whether they could act as attention getters to the gestures (as may be the case for ape multicomponent combinations; Line 96). But we did not find this effect. We have added this information in the aims of the study (See lines 214-218).

32.

What does "fairly stable" mean? Please give numbers.

Line 460-464: We now provide the Model Stability figures.

33.

Before submitting elsewhere, I would suggest moving the ethogram and the cumulative distribution of signal types to the ESM.

We moved the figure with the cumulative distribution of signal types to the Supplementary Information (Supplementary Figure 1), but we think that it is important to keep the ethogram (Table 2) in the main manuscript, so that future studies can refer to our descriptions and easily find videos of them in the ElephantVoices database.

34.

Table 3. "communications" is an odd category, especially for "Urinating".

We meant the number of greeting communications in which the olfactory behaviours were observed. We now clarified in lines 492-495.

35.

564: The authors make a big deal that 11/19 tactile signals were used when the recipient was visually attending, in contrast to 89% of "silent visual body acts." Yet, the authors also found that 83% of audible body contacts were used when the recipient was visually attending, although one would not predict this to be necessary. This latter result speaks against the author's hypothesis.

Thank you for raising this issue. Yes, in the section on the general frequency of use of body act modality according to the recipient's attentional state, we find that also audible body acts (namely Ear-Flapping and Ear-Slide) are used when recipients are visually attending (see Lines 512-514). With the new GLMM analysis we now also find that signallers use more audible and silent-visual gestures than tactile ones when recipients are attending (Lines 522-530). However, when we explored the percentage variation in use of body act modality according to recipient's visual attention (See lines 530-541), we show that silent-visual body acts decreased when recipients were not attending, while tactile body acts increased (a lot) when recipients were not attending and audible body acts also increased (although to a lesser degree) when recipients were not attending (Please see lines 398-406

and Supplementary Calculations 2 for more information on the calculations). So, we do show a shift in use, although slight in the use of audible gestures using the same methods as used in apes.

Our results are stronger than those reported previous results in apes, where more silent-visual gestures are used when recipients are attending and more tactile ones when they are not attending, whereas audible gestures were used irrespectively of the state of recipient visual attention. In contrast, even if slight, we do show an increase in the use of audible body acts when recipients are not attending (please see figure 4 in Hobaiter and Byrne 2011, from which we replicated our analyses).

It is important to note, as we mention in lines 297, that all gestures can be seen, so they are all “visual”. But, in addition to being visual, some involve contact with recipient (i.e, tactile) and some involve the intrinsic (rather than occasional or accidental) production of sounds (like our Ear-Flapping and Ear-Slide audible gestures). It is likely that Ear-Flapping and Ear-Slide (both audible gestures) provide both audible and visual information that is relevant to the recipient and are, thus, themselves “multimodal”. Moreover, because body acts are integrated with vocalisations during greeting, it may be that the audible information in the gesture is impacted by the vocalisation, making the visual information salient to its perception by the recipient. Future research studying use of gestures produced in the absence of vocalisations could explore whether audible gestures are used more often when recipients are not attending – but please note that the recipient’s visual attention is not relevant to their ability to receive audible information. We have clarified the interpretation of our results in lines 646-662.

36.

Figure 3 fails to acknowledge the small sample size for tactile signals. Moreover, the y-axis label is missing. Overall, this way of presenting the data in % deviation seems to be exaggerating the differences.

Thank you for raising this issue – we report our findings in this way to replicate previous work in apes, allowing for direct comparison of our findings. We now highlighted the small sample size by reporting it in the paragraph above in the figure legend and added the y axis. Please see the edited section (Lines: 398-406; Supplementary Information section “Additional information on calculations of adjustment of modality according to recipient visual attention”) and Supplementary Calculations 2 regarding our calculations to explore active adjustment of body act modality according to recipient visual attention (REF: Hobaiter & Byrne 2011).

37.

587: What are “tail body acts”?

We apologize for the confusion. They are body acts produced with the tail. We have now clarified the terminology. See line 306.

38.

What is the purpose of the analysis presented in Figure 5?

Thank you for your question. We wanted to further investigate the potential function of these tail body acts that don’t meet the audience directedness criterion. We looked at the distances at which all body acts are used at and found that, in contrast to other tail body acts, Tail-on-Side and Tail-Raise were produced within 1m from the recipient in the majority of cases, hinting that they may be gestures that may facilitate or invite access to the genitals by the recipient when sniffing with the trunk. We now moved these analyses on the possible functions of tail body acts in the Supplementary Information to shorten the manuscript (Section: “Additional analyses to further explore the function of tail body acts”).

39.

As mentioned above, the discussion is restricted to elephant communication, and the broader significance of the findings remains elusive.

Thank you for the suggestion. We have broadened the discussion to include other species' communication, including human language, and expanded on the significance of our findings (Lines Other species: Lines 619-624, 638-643, 652-656, 684-693, 716-718; Significance of findings: 734-754).

40.

802: delete the second "the."

Line 760: Thank you for noticing the typo. Now corrected.

Reviewer #2 (Remarks to the Author):

This paper seeks to increase our understanding of elephant communication by specifically investigating the use of multimodal signals in a semi-captive group during reunion events. Additionally, this paper seeks to add to the growing literature investigating the form, function, and evolution of multimodal communication more generally.

Overall, the paper is very thorough and well-written. The authors provide detailed descriptions of multimodal signals within the elephant communicatory repertoire, which bear recognition and further research. I have only a few suggestions-

Thank you very much for your positive feedback. We have addressed all your suggestions and comments in a revised version of the manuscript.

Title-

(1) I think "semi-captive" should be added to the title of the work, to indicate this work is done with a non-wild population.

We have now added "semi-captive" in the title as suggested.

Abstract-

(2) If possible, it would be nice to give a few details about the methods here (1-2 sentences, mentioning that these captured reunion episodes, and the types of observations/models that were used).

Thank you for your suggestion. We added the use of separation-reunion events and the models in lines 36-39 of the abstract.

Introduction -

(3) It might be helpful for the Introduction to be separated into sub-sections. There is a lot of really important information in here, covering a wide-range of topics. It would help orient the reader to have this information broken down into labeled sub-sections based on primary topics.

Thank you for your suggestion. We separated the introduction into sub-sections.

Methods-

(4) Study site and subjects- This sub-section might read better if you switched the order of the current first and second paragraph, such that lines 224-234 come first in this sub-section, followed by lines

215-223.

We edited as suggested (Lines 234-245).

Discussion-

(5) *It is unclear how this paper adds to our understanding of the evolution of multimodal communication (line 791-792, and also mentioned in the abstract) - perhaps the authors should add a sub-section to the discussion that covers this perspective more explicitly, or if the topic is not explicitly covered in this paper then perhaps this wording should be adjusted.*

Thank you for your suggestion. We now clarified this in the introduction and in the discussion conclusion (See Lines 45-49; 225-229; 733-745).

(6) *The authors seem to suggest that multimodal signals used by elephants during reunions act to strengthen bonds between individuals (lines 777-790). There are a number of papers about multimodal signal use for pair-bond maintenance and/or advertisement in the literature (including in various monogamous fish, birds, and primates; for example, see Singletary & Tecot (2020) "Multimodal pair-bond maintenance: A review of signaling across modalities in pair-bonded nonhuman primates"). While this area of research focuses largely on bonds between male-female mating pairs, it may be worth mentioning some of this work, and potentially making recommendations for future research that investigates this particular possible function (i.e., bond promotion for social rather than reproductive functions) in elephants.*

Thank you for your suggestion. We have added this information in the discussion (717-730).

(7) *It might be helpful to add a "Future Studies" sub-section to have some more explicit suggestions for where/how future research can be conducted to answer the remaining questions your paper introduces.*

Thank you for your suggestion. We added a conclusions and future directions in the conclusion (Lines 733-754).

(8) *It seems like there are a number of potential limitations to this study which are not discussed - this should be added as a "Limitations" sub-section.*

We address some limitations in the discussion and have now added an explicit (and expanded) conclusions and future directions section at the end of the discussion (Lines 731-752).

Reviewer #3 (Remarks to the Author):

The manuscript entitled "Multimodal communication in the greeting behaviour of African savannah elephants" describes a most interesting study on multimodal communication (greeting behaviour) in a population of semi-captive elephants. The authors collected an impressive amount of behaviours, and found evidence of first-order intentionality in elephant gestures.

The study is timely, the manuscript is well written and comprehensive, and it can be of interest for a wide audience. As such, I mostly have requests for clarifications or suggestions on how to improve the manuscript readability, together with a few more substantial requests. The latter concerns details about the methodology, and especially how the data were coded.

Thank you very much for your positive feedback on our study. We have addressed all your comments and have run inter-rater reliability on the coding.

Major points:

1.

• *In the introduction, the discussion on the greeting behaviour of wild elephants refers either the elephants separating and reuniting regularly (as a result of living in fission-fusion societies) or after periods of separation. While the experiment is based on separation and reunion procedure, I am wondering how comparable these situations are with the wild animals. More precisely: the separation in this experiment was rather short; do we expect differences in the greeting behaviour compared to much longer separations (eg, days?)*

Thank you for raising this issue. Our results are in line with descriptions of greetings between wild female elephants who may be separated for days or months as well as with those of a study on related captive female elephants that were reunited after years of separation (see Horner et al., 2021, *Animals*). So, it seems that elephants retain the same greeting behaviour in captivity and even for different periods of separation. We added the reference to this study in the introduction in lines 184-187 and in the discussion in lines 597-601. We were also surprised that elephants engaged in these highly excited greetings after short times of separation, but it seems to be genetically channeled in African savannah elephants and VE has now observed these excited greetings in the wild among female groups separated only for brief periods. We agree that future studies should explore in detail if the duration of separations affects the form of greetings.

2.

• *I missed in the methods several details on how the data were coded, especially considering that coding animal behaviour from videos can be a rather subjective endeavour:*

a) was there only one rater, or more than one?

Yes, there was only one rater.

b) If more than one, did the authors calculate inter-rater reliability or similar measures?

Thank you for raising this issue. We have now conducted inter-observer reliability calculating it for the variables of interest for this study: Signal_record (i.e., Body act record and Vocalisation record), Signaller Gaze, and Recipient Visual attention. We found a high level (>0.8 Kappa) of agreement for all the variables. We added this information in lines 340-345.

c) When coding greeting in a pair, was each animal considered simultaneously as a possible signaller and recipient?

Thank you for the question. Yes, if both elephants signalled at each other, we coded the greeting communications by the elephants in separate Elan files where we considered the first elephant signalling as “signaller” and the second elephant signalling as “recipient” in the first Elan file, named as ZEWACT_081121_Exp1_VEC0001a, and vice versa in the second elan file, named as ZEWACT_081121_Exp1_VEC0001b. We have added this information in the Supplementary Methods.

3.

• *Still in the methods: It is not clear to me how the distance during greeting was measured. Please provide more information (e.g., were there visible markers that the coders could use?)*

Thank you for this question. We estimated that the average body length of our elephants was 3m, so we counted the number of elephant lengths between the subjects as they were approaching and selected the different distance ranges (0-1, 1-3, 3-5 etc.). We have now added this information in lines 329-332.

Minor points:

4.

• *Lines 61 and following: the authors use sarcasm as an example of communication requiring both linguistic as well as paralinguistic cues to be deciphered. While I very much agree on this, I am not*

convinced that this is the best example for multi-modality: one can be sarcastic in writing, or even in face-to-face communication without adding gestures. I would suggest using a clearer example. I would also distinguish being sarcastic from making a joke, as these two terms can indicate rather different concepts.

Thank you for raising this issue. We removed the example about sarcasm and made a more general point about human language (Lines 59-63).

5.

• *Line 238: Could the authors add a few lines/give a few details in the manuscript to explain what the STRANGE framework is?*

We have added more information about the STRANGE framework in lines 240-245.

6.

• *Lines 264 and following: While I think it makes intuitive sense to selected animals showing strongest social bonds, I suggest to add a few lines to explain the rationale for this choice as, from the introduction, it wasn't clear to me that greeting behaviours were occurring more in socially closer animals.*

Thank you for raising this issue. We clarified in the introduction that the excited female greetings are of related or closely bonded females, and that captive elephants who are unfamiliar with each other do not greet, showing that social relationships impact greeting behaviour (See Lines 184-187 and in the discussion in lines 597-601).

7.

• *Line 547: "in 6% it was unknown whether elephants produced them" – I am not sure I understand the sentence. Please clarify.*

We edited to clarify that we were not sure the elephants defecated, urinated etc, if visibility was obscured in some way. See lines 492-495.

8.

• *Figure 3 and 4: Please reposition the labels "yes" and "no" on the x axis so that they are not inside the bars*

We have edited Figure 3 (now Figure 4) as suggested, and edited as suggested and moved Figure 4 into the Supplementary Information (Supplementary Figure 3).

9.

- *Line 612: The acronym MDCA was already explained, there is no need to repeat it here. We have removed the acronym.*

10.

- *Line 644: "Nearest-neighbour index affected the use of combinations of Rumble with Ear-flapping ($\chi^2=6.034$, $P=0.049$)."* – Where is this result reported? I cannot find it in Table 5. This is the result of the full-reduced model comparison (See details on all GLMM models now in Supplementary Information section: "Additional information on all GLMM models in the study"). It tests the overall effect of all fixed effects in the model and if it is non-significant then there is no need to explore the summary of the model. We added "overall" in lines 525, 562, 568 to clarify.

11.

- *Discussion, lines 793-795: The conclusion is rather vague, and could be improved by adding some examples of "promising venues" for future research or implications. Thank you for the suggestion. We have added these to the conclusion (see lines 746-754)*

Reviewers' comments:

Reviewer #1 (Remarks to the Author):

The authors invested great energy into the revision and rebuttal letter, and I thank them for their work. Nevertheless, I think some critical problems that require fixing remain.

Firstly, regarding the statistics, the authors acknowledge that their statistical analysis in one case violated the assumptions (line 381: "Because the body act types were produced by multiple signallers the assumption of independence of binomial tests was violated. An alternative test controlling for signaller could not be conducted due to the small sample sizes per body act type per signaller.") They then pressed ahead with the faulty binomial analysis anyway. If the sample size is too small for a meaningful and appropriate statistical analysis, the authors should just report the descriptive statistics.

Regarding the GLMM, the authors confused predictor and response variables. They write, "The response variable indicated whether the body act was produced when recipients were visually attending or not (Attending=1, Not attending=0) and the predictor variable indicated the signal modality of the body act (i.e., silent-visual, tactile, audible)." Yet, the response variable should be the type of signal produced and the predictor of whether or not the recipient is attending.

The statistical analyses need to be fixed before publication can be considered.

Secondly, the authors need to be more precise regarding the concept of intentionality. They need to distinguish between the different levels of intentionality from the outset (zero-order, first-order, second-order ...) and explain that the remarkable facet of human language and Theory of Mind is the connotation of second (and higher-)order intentionality, i.e., the ability to reason about and perhaps change someone else's mental state. First-order intentionality can be achieved via simple associative learning; it may also be evolved. Reporting first-order intentionality is nothing that requires advanced cognition. Indeed, many innate behaviours require the presence of 'releasers' – and one could conjure a situation where animals only produce facial expressions when they perceive others attending to them. This is not to say that elephant communication works this way. Still, the authors need to rule out such simpler accounts to make a case that the reported form of intentional communication goes beyond something that can be explained via associative learning or other simpler evolved mechanisms.

Third, the manuscript is still much too long. Is this manuscript based on a Master's thesis? The authors may want to consider shortening the manuscript considerably. We all have many things to read, and the authors are competing with others for readers' attention. It is more likely that people will read the paper in its entirety if it is well-motivated and engaging. The introduction and discussions have a word count of nearly 2000 words each and could easily be cut by half, specifically since they mainly focus on elephants. The manuscript has 180 references, but it's not a review paper. Again, I would strongly recommend paring down the references by about half.

Fourth, the data files submitted do not contain the critical signalling data. The files contain the computations of indices and summarized results but not the raw data or the code. I very strongly suggest that the authors make both data and code available to ensure reproducibility.

Minor comments:

Line 98: "Ideal" is perhaps a bit hyperbolic. Why not: "interesting" or "promising"?

Line 144: similar body plan within the order or with humans?

Line 320: "The communication" is an odd category – it includes diverse and non-mutually exclusive categories such as ID, context, and urinating. Perhaps rename it "communication event". That would also resolve later wording problems (e.g., line 392 ff: "Although multiple body acts occurred in the same communication we did not include communication as a random effect"). Are "communications" the same as „greeting communications" (line 469)?

543: explain „attraction“ when it is first used.

Reviewer #2 (Remarks to the Author):

This revised paper seeks to increase our understanding of elephant communication by specifically investigating the use of multimodal signals in a semi-captive group during reunion events. Additionally, this paper seeks to add to the growing literature investigating the form, function, and evolution of multimodal communication more generally.

The authors have clearly attended to the remarks of the three reviewers, and the paper seems much improved and does a better job of achieving its overall aims successfully. While there are some minor formatting issues throughout (i.e., inconsistency with formatting of headers, and issues in the reference list [for example, the first line mentions Zotero]), I have no remaining major remarks.

Reviewer #3 (Remarks to the Author):

I thank the authors for addressing my concerns.
I find the manuscript much improved since the previous version, and I approve of its publication.

Reviewers' comments:

Reviewer #1 (Remarks to the Author):

The authors invested great energy into the revision and rebuttal letter, and I thank them for their work. Nevertheless, I think some critical problems that require fixing remain.

Firstly, regarding the statistics, the authors acknowledge that their statistical analysis in one case violated the assumptions (line 381: "Because the body act types were produced by multiple signallers the assumption of independence of binomial tests was violated. An alternative test controlling for signaller could not be conducted due to the small sample sizes per body act type per signaller.") They then pressed ahead with the faulty binomial analysis anyway. If the sample size is too small for a meaningful and appropriate statistical analysis, the authors should just report the descriptive statistics.

Thank you for highlighting this issue. As you suggest we now removed the analysis and edited the methods and results sections (lines 299-307; 415-426). We also removed the results from the Descriptives tables in the Supplementary Material.

Regarding the GLMM, the authors confused predictor and response variables. They write, "The response variable indicated whether the body act was produced when recipients were visually attending or not (Attending=1, Not attending=0) and the predictor variable indicated the signal modality of the body act (i.e., silent-visual, tactile, audible)." Yet, the response variable should be the type of signal produced and the predictor of whether or not the recipient is attending.

Thank you for noticing this mistake. We now conducted a Multinomial logit model where we used body act modality as the response variable and recipient visual attention as predictor variable (Lines 307-314). We report the results including odds ratios and P. The results remain the same as our previous analysis that signallers use more silent visual or audible body acts as compared to tactile ones when greeting, showing audience directedness similarly to other studies on apes (see Lines 439-443). Thus, we have updated the statistics but our interpretation and discussion remain the same.

The statistical analyses need to be fixed before publication can be considered.

Secondly, the authors need to be more precise regarding the concept of intentionality. They need to distinguish between the different levels of intentionality from the outset (zero-order, first-order, second-order ...) and explain that the remarkable facet of human language and Theory of Mind is the connotation of second (and higher-)order intentionality, i.e., the ability to reason about and perhaps change someone else's mental state. First-order intentionality can be achieved via simple associative learning; it may also be evolved. Reporting first-order intentionality is nothing that requires advanced cognition. Indeed, many innate behaviours require the presence of 'releasers' – and one could conjure a situation where animals only produce facial expressions when they perceive others attending to them. This is not to say that elephant communication works this way. Still, the authors need to rule out such simpler accounts to make a case that the reported form of intentional communication goes beyond something that can be explained via associative learning or other simpler evolved mechanisms.

Thank you for your comment. We now realize that the confusion arises from us having used the word “intentional” and “first-order intentional” interchangeably. We now added more background on the different levels of intentionality, clarifying that the uniqueness of language relies on its second - (or higher-) order intentional use. For example Tomasello argues that: The uniqueness of language is believed to rely in its intentional production (Tomasello, 2005), and Dennett and Grice similarly outline that: with language we choose to communicate an underlying thought to a partner by taking into account their mental states (Dennett, 1983; Grice, 1969). We have clarified that we are exploring first-order intentionality by investigating audience directedness in the use of body acts by elephants and that first-order intentional use is an essential precursor to higher-order intentional use. Disentangling whether first-order intentional gestures are acquired via associative learning or evolved is beyond the scope of the manuscript. But, due to the high number of gestures showing audience directedness during greeting (and of potentially communicative body acts across elephant communication) we doubt that they are due to associative learning (e.g., the elephant sees the face of the other elephant towards him and produces the body act “Ear-Spread”). We believe, as you suggest, that it is rather an evolved mechanism for successful communication that allows animals to disentangle communicative gestures from body movements made without any communicative intent. While no study on apes has directly explored if first-order intentionality is acquired via associative learning, the presence of this capacity in even relatively young infant apes is suggestive of an evolved mechanism (as in humans). We believe that the most parsimonious assumption is that it is the same in elephants, a species with as complex society and cognition as apes (the first author is currently exploring first-order intentional gesturing across different contexts by different aged wild elephants and has seen cases of this ability even in young elephants). We do agree that this capacity does not require complex cognition, like an explicit theory of mind, and believe that it is probably widespread to other long-lived social species (e.g., cetaceans) that need to communicate social goals flexibly to navigate diverse relationships outside of kin. We believe that by showing this capacity across gestures of a contextual repertoire in a non-primate species, we are able to contribute new knowledge on close-range animal communication. (See edited lines 85-100; 526-533; 598-609).

Third, the manuscript is still much too long. Is this manuscript based on a Master's thesis? The authors may want to consider shortening the manuscript considerably. We all have many things to read, and the authors are competing with others for readers’ attention. It is more likely that people will read the paper in its entirety if it is well-motivated and engaging. The introduction and discussions have a word count of nearly 2000 words each and could easily be cut by half, specifically since they mainly focus on elephants. The manuscript has 180 references, but it’s not a review paper. Again, I would strongly recommend paring down the references by about half.

We have now cut the introduction to 1230 words, the Discussion to 1350 words, and the references by half.

Fourth, the data files submitted do not contain the critical signalling data. The files contain the computations of indices and summarized results but not the raw data or the code. I very strongly suggest that the authors make both data and code available to ensure reproducibility.

We provided all the code and data for data cleaning and statistical analyses (See lines 634; 639).

Minor comments:

Line 98: “Ideal” is perhaps a bit hyperbolic. Why not: “interesting” or “promising”?
Now changed to promising (line 72).

Line 144: similar body plan within the order or with humans?
Now changed to “share our body plan” (line 102).

Line 320: “The communication” is an odd category – it includes diverse and non-mutually exclusive categories such as ID, context, and urinating. Perhaps rename it “communication event”. That would also resolve later wording problems (e.g., line 392 ff: “Although multiple body acts occurred in the same communication we did not include communication as a random effect”). Are “communications” the same as „greeting communications“ (line 469)?
Now edited to “communication event” throughout.

543: explain „attraction“ when it is first used.
Line 334. Edited: “relative attraction (i.e., rate of co-occurrence)”

Reviewer #2 (Remarks to the Author):

This revised paper seeks to increase our understanding of elephant communication by specifically investigating the use of multimodal signals in a semi-captive group during reunion events. Additionally, this paper seeks to add to the growing literature investigating the form, function, and evolution of multimodal communication more generally.

The authors have clearly attended to the remarks of the three reviewers, and the paper seems much improved and does a better job of achieving its overall aims successfully. While there are some minor formatting issues throughout (i.e., inconsistency with formatting of headers, and issues in the reference list [for example, the first line mentions Zotero]), I have no remaining major remarks.

We thank Reviewer 2 for finding the manuscript now suitable for publication. We addressed the formatting issues.

Reviewer #3 (Remarks to the Author):

I thank the authors for addressing my concerns.
I find the manuscript much improved since the previous version, and I approve of its

publication.

We thank Reviewer 3 for considering the manuscript now suitable for publication